# Ccrk-Mak/Ick signaling is a ciliary transport regulator essential for retinal photoreceptor survival

Taro Chaya[1],*, Yamato Maeda[1],*, Ryotaro Tsutsumi[1], Makoto Ando[1], Yujie Ma[1], Naoko Kajimura[2] ⓘ, Teruyuki Tanaka[3], Takahisa Furukawa[1] ⓘ

**Primary cilia are microtubule-based sensory organelles whose dysfunction causes ciliopathies in humans. The formation, function, and maintenance of primary cilia depend crucially on intraflagellar transport (IFT); however, the regulatory mechanisms of IFT at ciliary tips are poorly understood. Here, we identified that the ciliopathy kinase Mak is a ciliary tip-localized IFT regulator that cooperatively acts with the ciliopathy kinase Ick, an IFT regulator. Simultaneous disruption of _Mak_ and _Ick_ resulted in loss of photoreceptor ciliary axonemes and severe retinal degeneration. Gene delivery of _Ick_ and pharmacological inhibition of FGF receptors, Ick negative regulators, ameliorated retinal degeneration in _Mak_$^{-/-}$ mice. We also identified that Ccrk kinase is an upstream activator of Mak and Ick in retinal photoreceptor cells. Furthermore, the overexpression of Mak, Ick, and Ccrk and pharmacological inhibition of FGF receptors suppressed ciliopathy-related phenotypes caused by cytoplasmic dynein inhibition in cultured cells. Collectively, our results show that the Ccrk-Mak/Ick axis is an IFT regulator essential for retinal photoreceptor maintenance and present activation of Ick as a potential therapeutic approach for retinitis pigmentosa caused by _MAK_ mutations.**

## Introduction

Primary cilia are hair-like microtubule-based structures that extend from the basal bodies of almost all cell types, and play key roles in a variety of sensory functions across species (1, 2). A diverse range of signaling receptors, ion channels, and their downstream effectors localized to the primary cilia sense and decode extracellular stimuli, including light and Hedgehog morphogens (3). For example, retinal photoreceptor cells possess a light-sensory structure containing components of the phototransduction cascade, including Rhodopsin and cone opsins, the outer segments, which are the specialized primary cilia that have the convoluted membranous-associated axonemes but share common structural features with those of primary cilia in other cell types (4). Therefore, cilia are recognized as the signaling centers for multiple transduction pathways. In humans, ciliary dysfunction causes diseases known as ciliopathies, which are characterized by a broad spectrum of pathologies, including polydactyly, craniofacial abnormalities, brain malformation, intellectual disability, obesity, diabetes, polycystic kidney disease, anosmia, hearing loss, and retinal degeneration (5, 6, 7). Although many causative genes for ciliopathies have been identified, how genetic changes lead to the clinical phenotypes of ciliopathies with variable expressivity and severity is largely unknown (8).

The formation, length control, and function of cilia rely on intraflagellar transport (IFT), a bidirectional protein transport system coordinated by three large protein complexes, IFT-B, IFT-A, and BBSome, with molecular motors along the ciliary axonemal microtubules. They assemble into highly repetitive polymers called IFT trains, which move ciliary cargoes along the axoneme in both anterograde and retrograde directions (9, 10, 11). The three IFT complexes also function in the import and export of ciliary proteins (12, 13). IFT-B mediates anterograde transport from the base to the tip of the cilium driven by the kinesin-2 motor, whereas IFT-A mediates retrograde transport from the tip back to the base powered by the cytoplasmic dynein-2 motor (9, 12). The BBSome acts as a cargo adaptor that mediates the exit of membrane proteins from cilia (12, 14, 15). Recent studies have revealed detailed 3D structures of IFT-B, IFT-A, and BBSome (16, 17, 18, 19, 20, 21, 22, 23, 24, 25). Mutations in genes encoding IFT components are known to cause ciliopathies, including the Bardet–Biedl syndrome (BBS) (7). At the tip of the cilia, IFT trains disassemble and reassemble for turnaround and retrograde transport (26); however, the underlying regulatory mechanisms remain poorly understood (27). Sensory

---

[1]Laboratory for Molecular and Developmental Biology, Institute for Protein Research, Osaka University, Osaka, Japan  [2]Research Center for Ultra-High Voltage Electron Microscopy, Osaka University, Osaka, Japan  [3]Department of Developmental Medical Sciences, Graduate School of Medicine, The University of Tokyo, Tokyo, Japan

Correspondence: takahisa.furukawa@protein.osaka-u.ac.jp
Ryotaro Tsutsumi's present address is Department of Pathological and Health Science, School of Pharmaceutical Sciences, Wakayama Medical University, Wakayama, Japan
Teruyuki Tanaka's present address is Department of Pediatrics, Tokyo Children's Rehabilitation Hospital, Tokyo, Japan
*Taro Chaya and Yamato Maeda contributed equally to this work

cilia that have complex membrane-bounded axonemes similar to vertebrate photoreceptors, as well as the presence of extensive distal singlet microtubule domains associated with ciliary signaling, are found in the *Caenorhabditis elegans* (*C. elegans*) nervous system. There is significant system-specific variability in the details of IFT; for example, in contrast to *Chlamydomonas reinhardtii* (*Chlamydomonas*) and some mammalian cilia, two distinct anterograde motors operate in *C. elegans* cilia (28) and the return of kinesin-2 motors from the ciliary tip to the base occurs through diffusion in *Chlamydomonas* (26, 29), by retrograde transport in *C. elegans*, and by both processes in mammalian cells (28, 30, 31). Similarly, the turnaround mechanism also appears to be different in different systems depending on whether only one or two distinct anterograde motors are involved (28, 32).

We previously identified intestinal cell kinase (Ick), also known as ciliogenesis-associated kinase 1 (Cilk1), as a regulator of IFT turnaround at the ciliary tip (33, 34). *Chlamydomonas* LF4, *Tetrahymena* LF4A, *Leishmania mexicana* LmxMPK9, and *C. elegans* DYF-5, orthologues of Ick, are also reported to be involved in the regulation of IFT, as well as ciliary/flagellar length and formation (35, 36, 37, 38, 39, 40). *Ick* deficiency leads to dysregulation of ciliary length, impaired Hedgehog signaling, and the accumulation of IFT-B, IFT-A, and BBSome components at the tips of cilia, whereas Ick overexpression induces the accumulation of IFT-B, but not IFT-A or BBSome components at ciliary tips (33, 41, 42, 43, 44). *Ick*-deficient mice exhibit neonatal lethality, with developmental abnormalities observed in multiple organs and tissues, including the bone, lung, kidney, and inner ear (45). In humans, homozygous loss-of-function mutations in the *ICK* gene cause endocrine-cerebro-osteodysplasia syndrome, an autosomal recessive ciliopathy showing neonatal lethality with various developmental defects involving the endocrine, cerebral, and skeletal systems, and short rib-polydactyly syndrome, an autosomal recessive ciliopathy characterized by perinatal lethality with short ribs, shortened and hypoplastic long bones, polydactyly, and multiorgan system abnormalities (46, 47, 48). In addition, heterozygous variants of the human *ICK* gene are associated with juvenile myoclonic epilepsy (49). Collectively, Ick is currently recognized as a critical regulator of the IFT turnaround step at the ciliary tips (16, 27).

In the present study, we screened and found that male germ cell–associated kinase (Mak) regulates the IFT turnaround step at the ciliary tip. Mouse models revealed that *Mak* and *Ick* genetically interact in IFT regulation and play a crucial role in retinal photoreceptor maintenance, although *Ick* had a minor role. Mak overexpression restored ciliary defects in *Ick*-deficient cultured cells. Gene delivery of *Ick* and pharmacological inhibition of FGF receptors, negative regulators of Ick, ameliorated retinal degeneration in $Mak^{-/-}$ mice, a retinitis pigmentosa model. In addition, cell cycle–related kinase (Ccrk) was identified as a critical upstream regulator of Mak/Ick in retinal photoreceptor cells. Furthermore, the overexpression of Mak, Ick, and Ccrk and pharmacological inhibition of FGF receptors rescued the ciliopathy-related phenotypes resulting from cytoplasmic dynein inhibition. Taken together, this study identified a kinase signaling pathway regulating the IFT that plays an essential role in retinal photoreceptor maintenance.

## Results

### Ick plays a minor role in the regulation of IFT in retinal photoreceptor cells

To investigate the role of Ick in retinal photoreceptor development and maintenance, we mated *Ick* flox mice (33) with *Dkk3-Cre* mice (50), which predominantly express Cre recombinase in retinal progenitor cells, generated *Ick* conditional knockout (CKO) mice, and first analyzed *Ick* CKO mice at 1 mo. To observe the subcellular localization of Ick in retinal photoreceptor cells, immunohistochemical analysis was performed using an anti-Ick antibody. We found Ick signals in the distal regions of the photoreceptor ciliary axonemes in the control retina but not in the *Ick* CKO retina (Fig S1A). We also found signals stained with the anti-Ick antibody in the inner segments and outer nuclear layer (ONL) in both control and *Ick* CKO retinas (Fig S1A), suggesting that these signals were nonspecific. To examine whether *Ick* deficiency affects the distribution of IFT components in photoreceptor cilia, we immunostained retinal sections using anti-IFT88 (an IFT-B component), anti-IFT140 (an IFT-A component), and anti-acetylated α-tubulin (Actub, a ciliary axoneme marker) antibodies, and observed no substantial differences between the control and *Ick* CKO retinas (Fig S1B). We also performed immunohistochemical analyses using marker antibodies against Rhodopsin (rod outer segments), S-opsin (S-cone outer segments), and M-opsin (M-cone outer segments) and observed no substantial differences between the control and *Ick* CKO rod and cone outer segments (Fig S1C). To evaluate the electrophysiological properties of the *Ick* CKO retina, electroretinograms (ERGs) of *Ick* CKO mice were measured under dark-adapted (scotopic) and light-adapted (photopic) conditions. Under scotopic conditions, the amplitude of a-waves and b-waves, originating mainly from the population activity of rod photoreceptor cells (a-waves) and rod bipolar cells (b-waves), was unaltered between the control and *Ick* CKO mice (Fig S1D and E). Similar to scotopic ERG, the amplitudes of photopic a-waves and b-waves, which mainly reflect the population activity of cone photoreceptor cells (a-waves) and cone ON bipolar cells (b-waves), were comparable between the control and *Ick* CKO mice (Fig S1D and E).

To investigate the effects of *Ick* deficiency on retinal photoreceptor cells at later stages, we performed histological analyses using retinal sections from *Ick* CKO mice at 6 mo. Toluidine blue staining showed that the ONL thickness decreased in the *Ick* CKO retina compared with that in the control retina from postnatal day 14 (P14), probably because of the reduced proliferation of retinal progenitor cells (Fig S2A and B) (33). In contrast, the proportion of the ONL thickness to the inner retinal layer thickness decreased in *Ick* CKO mice compared with that in the control mice at 6 mo, but not at P14 or 1 mo, indicating that the ONL thickness decreased more in *Ick* CKO mice compared with that in the control mice at 6 mo (Fig S2C). Immunohistochemical examination using marker antibodies against Rhodopsin, S-opsin, and M-opsin showed mislocalization of M-opsin in the inner part of the photoreceptor cells in the *Ick* CKO retina, although there were no obvious differences in the Rhodopsin and S-opsin signals between the control and *Ick* CKO retinas (Fig S2D). We also performed an ERG analysis

and found that scotopic a- and b-waves and photopic b-wave amplitudes in *Ick* CKO mice were lower than those in the control mice (Fig S2E and F). These results suggest slowly progressive retinal degeneration in *Ick* CKO mice and a minor role of Ick in regulating IFT in retinal photoreceptor cells.

## Mak plays a major role in the regulation of IFT in retinal photoreceptor cells

Based on the above results, we hypothesized that serine–threonine kinase(s) other than Ick function as IFT regulators in the retinal photoreceptor cells. We focused on nine serine–threonine kinases close to Ick in the phylogenetic tree of the human kinome (51) and examined their subcellular localization using FLAG-tagged constructs (Fig 1A). We observed that Mak and cyclin-dependent kinase-like 5 (Cdkl5), as well as Ick, localized to ciliary tips in cultured cells (Fig 1B). To test whether the overexpression of Mak and Cdkl5 affects IFT, we transfected plasmids encoding FLAG-tagged IFT57 (an IFT-B component) into NIH3T3 cells with the Mak- or Cdkl5-expressing construct (Fig 1A). IFT57 was distributed along the ciliary axoneme without the overexpression of Mak or Cdkl5. Similar to Ick, Mak, but not Cdkl5, overexpression induced IFT57 accumulation at the ciliary tips (Figs 1C and D and S3A), suggesting that Mak, rather than Cdkl5, is a candidate IFT regulator. To investigate the functional roles of Cdkl5 in retinal photoreceptor cells, we examined the tissue distribution of *Cdkl5* transcripts by RT–PCR analysis using mouse tissue cDNAs at 4 wk of age and observed that *Cdkl5* is ubiquitously expressed in various tissues, including the retina (Fig S3B). To evaluate the effects of *Cdkl5* deficiency on retinal function, we performed an ERG analysis and found no obvious differences in the amplitudes of scotopic and photopic a- and b-waves between $Cdkl5^{+/Y}$ and $Cdkl5^{-/Y}$ mice at 1 and 3 mo (Fig S3C), suggesting that *Cdkl5* is dispensable for retinal photoreceptor function and maintenance. In contrast, we previously reported that *Mak* is highly expressed in retinal photoreceptor cells and that $Mak^{-/-}$ mice exhibit progressive retinal degeneration (52, 53). Subsequent analyses showed that mutations in the human *MAK* gene cause the retinal degenerative disease retinitis pigmentosa (54, 55). Together, our results and those of previous reports imply that Mak, rather than Cdkl5, functions as an IFT regulator in retinal photoreceptor cells.

To assess the role of Mak in IFT in detail, we transfected plasmids encoding FLAG-tagged IFT88, IFT140, or BBS8 (a BBSome component) into NIH3T3 cells with or without the Mak-expressing construct. IFT88 was distributed along the ciliary axoneme in the absence of Mak overexpression. Under these conditions, IFT140 and BBS8 were localized mainly to the ciliary bases. Similar to Ick, Mak overexpression induced accumulation of IFT88, but not IFT140 or BBS8, at the ciliary tips (Figs 1E and F and S3D and E). The IFT140 signals at the ciliary bases increased in Mak- and Ick-overexpressing cells (Fig S3D). We also observed the accumulation of IFT57 at the ciliary tips in human MAK- and ICK-overexpressing cells (Fig S3F and G). To examine whether *Mak* deficiency affects the distribution of IFT components in retinal photoreceptor cilia, we immunostained retinal sections from $Mak^{-/-}$ mice at 1 mo using anti-IFT88, anti-IFT140, and anti-Actub antibodies. As observed in our previous study (52), photoreceptor connecting cilia were elongated in the

$Mak^{-/-}$ retina compared with those in the $Mak^{+/+}$ retina (Fig 1G). In contrast to the *Ick* CKO retinas (Fig S1B), we observed that both IFT88 and IFT140 were concentrated at the tips of photoreceptor connecting cilia in the $Mak^{-/-}$ retina compared with those in the $Mak^{+/+}$ retina (Fig 1G), which is similar to our previous observation that IFT components are concentrated at the ciliary tips in $Ick^{-/-}$ MEFs (33). This observation shows that *Mak* deficiency, as well as *Ick* deficiency, impairs retrograde IFT. It has been previously reported that Ick phosphorylates Kif3a, a subunit of kinesin-2 (33, 56). To test whether Mak can also phosphorylate Kif3a, we performed a Phos-tag Western blot analysis and observed a decrease in the up-shifted band of Kif3a in the $Mak^{-/-}$ retina compared with that in the $Mak^{+/+}$ retina, suggesting that Kif3a is phosphorylated by Mak in the retina (Fig 1H). Together, these results suggest that Mak plays a major role in the regulation of IFT in retinal photoreceptor cells compared with Ick.

## Disruption of both *Mak* and *Ick* causes severe progressive retinal degeneration

To further investigate the functional roles of Mak and Ick in retinal photoreceptor cells, we generated *Mak Ick* double knockout (DKO) mice by mating $Mak^{-/-}$ mice with *Ick* CKO mice. We first performed histological analyses using retinal sections from *Mak Ick* DKO mice at P14 and 1 mo. Toluidine blue staining showed that the ONL thickness progressively decreased in $Mak^{-/-}$ and *Mak Ick* DKO retinas compared with that in the control retina and that the extent of the decrease in the *Mak Ick* DKO retina was greater than that in the $Mak^{-/-}$ retina (Fig 2A and B). Immunohistochemical examination using marker antibodies showed mislocalization of Rhodopsin, S-opsin, and M-opsin in the inner part of retinal photoreceptor cells in $Mak^{-/-}$ and *Mak Ick* DKO mice, as well as severe mislocalization in *Mak Ick* DKO mice compared with that in $Mak^{-/-}$ mice (Figs 2C and S4A). We did not observe rod and cone outer segment structures in *Mak Ick* DKO mice (Figs 2C and S4A). To examine whether deficiency of both *Mak* and *Ick* affects ciliary formation and the distribution of IFT components in the cilia of retinal photoreceptor cells, we immunostained retinal sections from *Mak Ick* DKO mice at P9 and P14 using antibodies against Actub, γ-tubulin (γTub, a basal body marker), pericentrin (a basal body marker), and IFT88. Notably, photoreceptor ciliary axonemes were not observed in the *Mak Ick* DKO retina, although those in the $Mak^{-/-}$ retina were elongated (Figs 2D and S4B). We observed that IFT88 signals were concentrated near the basal bodies of retinal photoreceptor cells in *Mak Ick* DKO mice (Fig 2D). To observe the ultrastructure of retinal photoreceptor cilia, we performed an electron microscopic analysis. Although we found basal bodies, connecting cilia and outer segments were not observed in the retina of *Mak Ick* DKO mice at P14, which is consistent with the results of immunohistochemical analysis (Fig 2E and F). To evaluate the electrophysiological properties of the *Mak Ick* DKO retina, we performed an ERG analysis and found that scotopic and photopic a- and b-wave amplitudes in $Mak^{-/-}$ mice were lower than those in the control mice (Fig 2G and H). In *Mak Ick* DKO mice, no significant ERG responses were detected (Fig 2G and H). These results suggest that deletion of both *Mak* and *Ick* causes severe retinal degeneration compared with deletion of

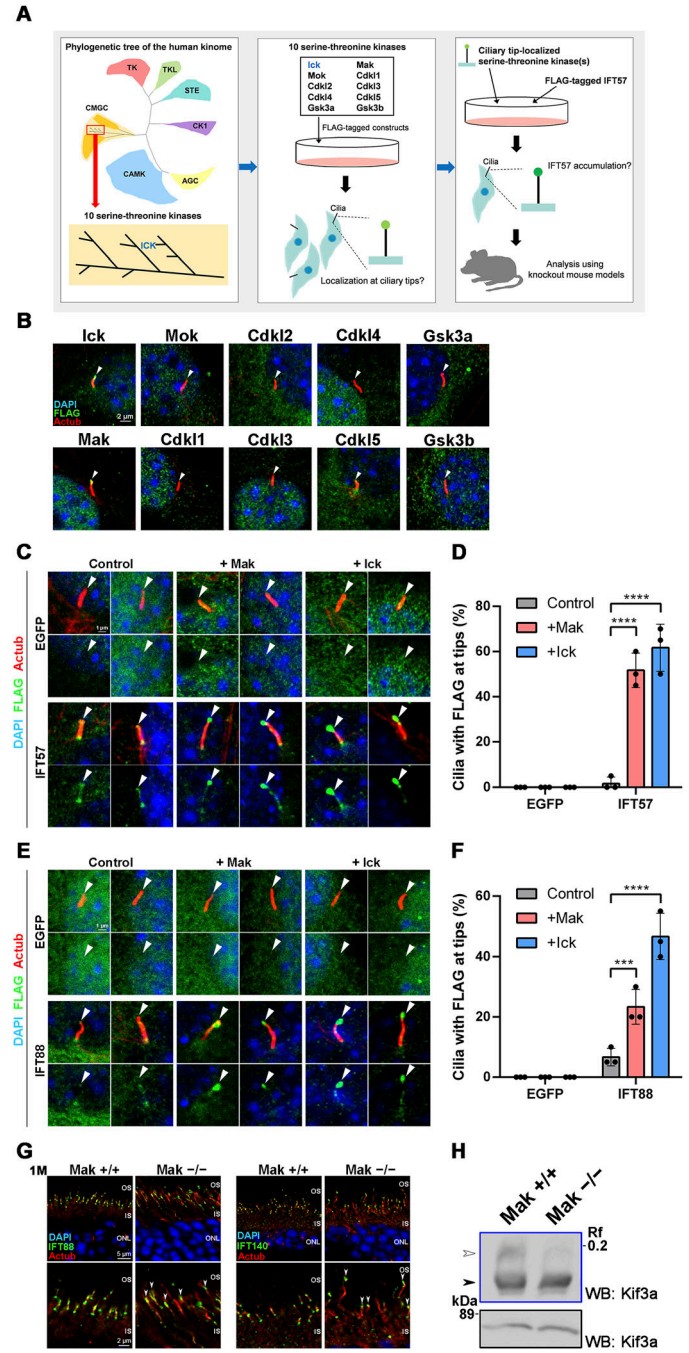

**Figure 1. Ciliary localization of IFT components is changed by Mak overexpression or knockout.**
**(A)** Schematic diagrams of the experimental workflow for the screening of serine–threonine kinase(s) other than Ick that function as IFT regulators in retinal photoreceptor cells. **(B)** Subcellular localization of Ick and nine serine–threonine kinases that are close to Ick in the phylogenetic tree of the human kinome. A plasmid encoding a FLAG-tagged Ick, Mak, Mok, Cdkl1, Cdkl2, Cdkl3, Cdkl4, Cdkl5, Gsk3a, or Gsk3b was transfected into NIH3T3 cells. Cells were immunostained with anti-FLAG and anti-Actub antibodies. Arrowheads indicate ciliary tips. **(C, D)** Effects of Mak and Ick overexpression on ciliary localization of IFT57. **(C)** FLAG-tagged EGFP or IFT57 expression plasmid was co-transfected into NIH3T3 cells with a plasmid expressing Mak or Ick. Cells were immunostained with anti-FLAG and anti-Actub antibodies. Arrowheads indicate ciliary tips. **(D)** Number of cilia with the FLAG-tagged EGFP or IFT57 predominantly localized at ciliary tips was counted. Data are presented as the mean ± SD. ****$P < 0.0001$ (two-way ANOVA followed by Tukey's multiple comparisons test). n = 3 experiments. **(E, F)** Effects of Mak and Ick overexpression on ciliary localization of IFT88. **(E)** FLAG-tagged EGFP or IFT88 expression plasmid was co-transfected into NIH3T3 cells with a plasmid expressing Mak or Ick. Cells were immunostained with anti-FLAG and anti-Actub antibodies. Arrowheads indicate ciliary tips. **(F)** Number of cilia with the FLAG-tagged EGFP or IFT88 predominantly localized at ciliary tips was counted. Data are presented as the mean ± SD. ***$P < 0.001$, ****$P < 0.0001$ (two-way ANOVA followed by Tukey's multiple comparisons test). n = 3 experiments. **(G)** Ciliary localization of IFT components in photoreceptor cells of the $Mak^{-/-}$ mouse retina. Retinal sections obtained from $Mak^{+/+}$ and $Mak^{-/-}$ mice at 1 mo were immunostained using antibodies against IFT88, IFT140, and Actub. IFT components were concentrated at the tips of photoreceptor connecting cilia in the $Mak^{-/-}$ retina (arrowheads). OS, outer segment; IS, inner segment; ONL, outer nuclear layer. **(H)** Kif3a phosphorylation in the $Mak^{+/+}$ and $Mak^{-/-}$ retina. The retinal lysates were analyzed by Phos-tag

*Mak* alone and that *Mak* and *Ick* genetically interact and play a central role in the regulation of IFT in retinal photoreceptor cells.

### Heterozygous deletion of *Ick* exacerbates retinal degeneration in *Mak⁻/⁻* mice

Previous reports have shown that loss-of-function mutations and variants are present in the human *ICK* gene ([46], [47], [48], [49]). To gain insight into the effects of heterozygous *ICK* mutations or variants on the symptoms of retinitis pigmentosa caused by homozygous mutations in the *MAK* gene in humans, we sought to generate and analyze *Mak⁻/⁻*; *Ick⁺/⁻* mice. We analyzed a publicly available single-cell RNA-sequencing (scRNA-seq) dataset of adult human retinas ([57]). In total, 20 cell clusters were identified using a principal component analysis (PCA)–based approach and projected by a Uniform Manifold Approximation and Projection (UMAP) onto a two-dimensional plot ([Fig 3A]). We observed the expression of *MAK* and *ICK* in photoreceptor cells of the human retina ([Fig 3A and B]). We confirmed that *MAK* and *ICK* are expressed in the human retina by RT–PCR analysis ([Fig S5A]). To investigate the effects of *Ick* heterozygous deficiency on retinal photoreceptor cells, we performed histological analyses using retinal sections from *Mak⁺/⁻*; *Ick⁺/⁻* mice at 6 mo. We previously observed no obvious retinal structural differences between WT and *Mak⁺/⁻* mice at 6 mo ([52]). Toluidine blue staining showed no significant differences in the ONL thickness between the *Mak⁺/⁺*; *Ick⁺/⁺* and *Mak⁺/⁻*; *Ick⁺/⁻* retinas ([Fig S5B and C]). Immunohistochemical examination using marker antibodies against Rhodopsin, S-opsin, and M-opsin showed no obvious differences between the *Mak⁺/⁺*; *Ick⁺/⁺* and *Mak⁺/⁻*; *Ick⁺/⁻* retinas ([Fig S5D]). To evaluate the effects of *Ick* heterozygous deficiency on retinal function, we performed an ERG analysis and found no significant differences in the amplitudes of scotopic and photopic a- and b-waves between *Mak⁺/⁺*; *Ick⁺/⁺* and *Mak⁺/⁻*; *Ick⁺/⁻* mice ([Fig S5E and F]). These results show that *Ick* heterozygous deficiency alone does not affect retinal photoreceptor function or maintenance.

Next, we performed histological analyses of retinal sections from *Mak⁻/⁻*; *Ick⁺/⁻* mice at 2 mo. Toluidine blue staining showed that the ONL thickness decreased in *Mak⁻/⁻* and *Mak⁻/⁻*; *Ick⁺/⁻* retinas compared with that in the *Mak⁺/⁻* retina and that the extent of the decrease in the *Mak⁻/⁻*; *Ick⁺/⁻* retina was greater than that in the *Mak⁻/⁻* retina ([Fig 3C and D]). Immunohistochemical examination using marker antibodies showed disorganized structures of the rod and cone outer segments in *Mak⁻/⁻* and *Mak⁻/⁻*; *Ick⁺/⁻* mice, and severe disorganization in *Mak⁻/⁻*; *Ick⁺/⁻* mice compared with that in *Mak⁻/⁻* mice ([Fig 3E]). The rod outer segment length decreased in the *Mak⁻/⁻*; *Ick⁺/⁻* retina compared with that in the *Mak⁻/⁻* retina ([Fig 3E and F]). To evaluate the electrophysiological properties of the *Mak⁻/⁻*; *Ick⁺/⁻* retina, we performed an ERG analysis and found that scotopic and photopic a- and b-wave amplitudes in *Mak⁻/⁻* and *Mak⁻/⁻*; *Ick⁺/⁻* mice were lower than those in *Mak⁺/⁻* mice and that the amplitudes of scotopic a-waves and photopic a- and b-waves significantly decreased in *Mak⁻/⁻*; *Ick⁺/⁻* mice compared with those in *Mak⁻/⁻* mice ([Fig 3G and H]). These results suggest that *ICK* can be

a modifier of retinitis pigmentosa caused by *MAK* mutations in humans.

### The abnormalities caused by *Ick* and *Mak* deficiency are rescued by activation of *Mak* and *Ick*, respectively

To examine whether ciliary defects caused by *Ick* deficiency can be restored by Mak activation, we transfected plasmids encoding Mak into *Ick⁻/⁻* MEFs. We previously reported that ciliated cells are fewer and ciliary length is shorter in *Ick⁻/⁻* MEFs than in *Ick⁺/⁺* MEFs ([33]). Remarkably, the overexpression of Mak, similar to that of Ick, rescued the ciliary defects observed in *Ick⁻/⁻* MEFs ([Fig 4A–C]). To investigate whether retinal abnormalities caused by *Mak* deficiency can also be restored by Ick activation, we first injected an adeno-associated virus (AAV) expressing Ick under the control of the Rhodopsin kinase promoter ([58]) into the *Mak⁻/⁻* retina ([Fig 4D]). We observed that the mislocalization of Rhodopsin to the inner part of rod photoreceptor cells in the *Mak⁻/⁻* retina was partially suppressed by Ick expression in retinal photoreceptor cells ([Fig 4E and F]). It was previously reported that fibroblast growth factor receptors (Fgfrs) negatively regulate Ick activity ([59]). Next, we injected BGJ398, an FDA-approved small-molecule inhibitor of Fgfrs, into *Mak⁻/⁻* mice to activate Ick in retinal photoreceptor cells ([Fig 4G]). Toluidine blue staining showed an increase in the ONL thickness of the retina in BGJ398-treated *Mak⁻/⁻* mice compared with that in vehicle-treated *Mak⁻/⁻* mice ([Fig 4H and I]). To evaluate the effects of BGJ398 on retinal function in *Mak⁻/⁻* mice, we performed an ERG analysis and observed that scotopic a- and b-wave amplitudes in BGJ398-treated *Mak⁻/⁻* mice were higher than those in vehicle-treated *Mak⁻/⁻* mice ([Fig 4J]). To test the effects of BGJ398 on the activity of Fgfrs in the retina, we performed a Phos-tag Western blot analysis and observed that the Ick substrate Kif3a was more phosphorylated in the retina of BGJ398-treated *Mak⁻/⁻* mice than in that of vehicle-treated *Mak⁻/⁻* mice, suggesting that Ick is activated by BGJ398 in the *Mak⁻/⁻* retina ([Fig 4K and L]). To examine whether BGJ398-mediated rescue of the defects in the *Mak⁻/⁻* retina is through Ick, we injected BGJ398 into *Mak Ick* DKO mice ([Fig S6A]). In contrast to the *Mak⁻/⁻* retina, toluidine blue staining showed no significant differences in the ONL thickness of the retinas between vehicle- and BGJ398-treated *Mak Ick* DKO mice ([Fig S6B and C]). To evaluate the effects of BGJ398 on retinal function in *Mak Ick* DKO mice, we performed an ERG analysis and observed no obvious differences in scotopic a- and b-wave amplitudes between vehicle- and BGJ398-treated *Mak Ick* DKO mice ([Fig S6D]). These data from BGJ398-treated *Mak Ick* DKO mice support the idea that the rescue effects of BGJ398 on retinal degeneration in *Mak⁻/⁻* mice are mediated by Ick. To test the effects of BGJ398 on WT retinas, we injected BGJ398 into *Mak⁺/⁺* mice and performed histological and electrophysiological analyses ([Fig S6A]). In contrast to *Mak⁻/⁻* mice, we observed no significant differences in the ONL thickness or scotopic ERG a- and b-wave amplitudes between vehicle- and BGJ398-treated *Mak⁺/⁺* mice ([Fig S6E–G]). Collectively, these results show

(blue box) and conventional (black box) Western blotting using an anti-Kif3a antibody. The level of Kif3a phosphorylation presented as the ratio of the upper band intensity (white arrowhead) to the lower band intensity (black arrowhead) decreased in the *Mak⁻/⁻* retina compared with that in the *Mak⁺/⁺* retina. Nuclei were stained with DAPI.

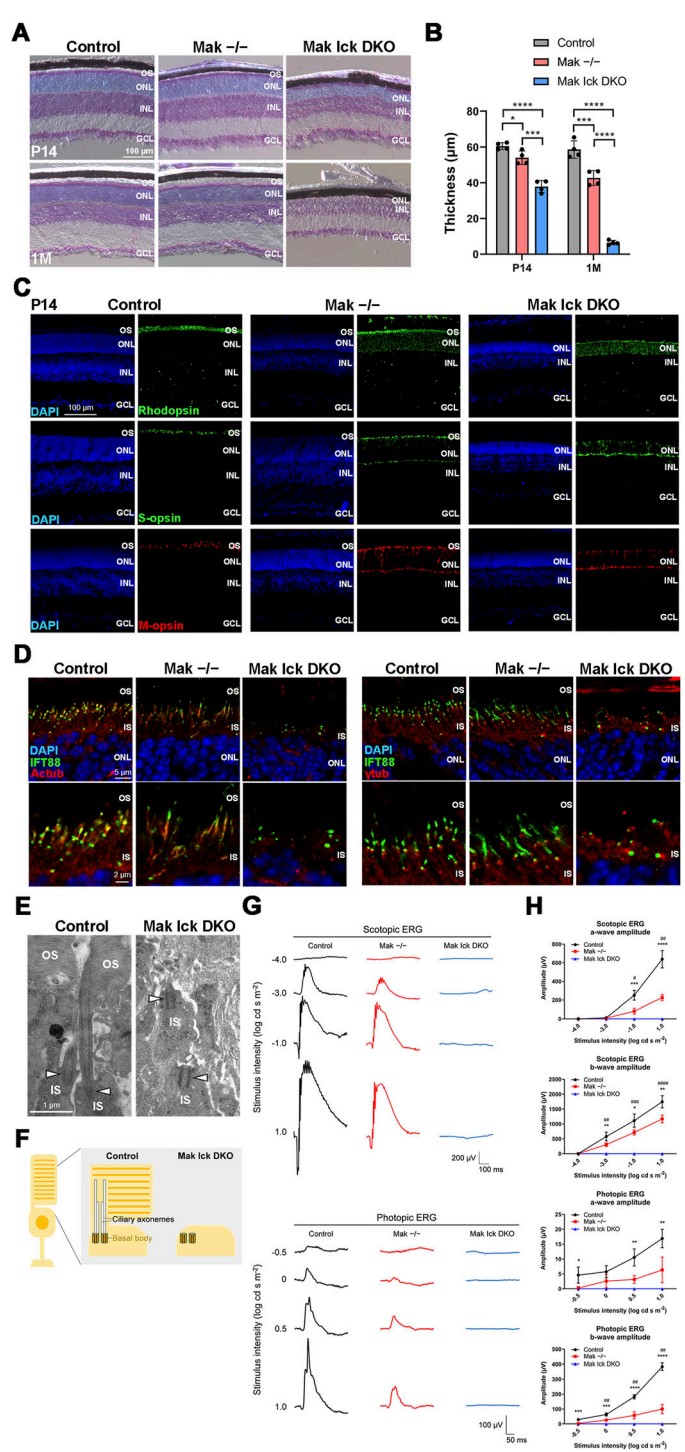

**Figure 2. Severe progressive retinal degeneration in *Mak Ick* DKO mice.**

**(A, B)** Toluidine blue staining of retinal sections from the control, *Mak*−/−, and *Mak Ick* DKO mice at P14 and 1 mo. The ONL thickness was measured. Data are presented as the mean ± SD. *P < 0.05, ***P < 0.001, ****P < 0.0001 (one-way ANOVA followed by Tukey's multiple comparisons test). n = 4 mice per each genotype. **(C)** Immunostaining of retinal sections from the control, *Mak*−/−, and *Mak Ick* DKO mice at P14 using marker antibodies against Rhodopsin, S-opsin, and M-opsin. Severe photoreceptor outer segment disorganization and mislocalization of Rhodopsin and cone opsins were observed in the *Mak Ick* DKO retina. **(D)** Ciliary localization of IFT components in photoreceptor cells of the *Mak Ick* DKO retina. Retinal sections obtained from the control, *Mak*−/−, and *Mak Ick* DKO mice at P14 were immunostained using antibodies against IFT88, Actub, and γ-tubulin (γTub) (a marker for basal bodies). The ciliary axoneme of retinal photoreceptor cells was absent in *Mak Ick* DKO mice. **(E)** Longitudinal profiles of the connecting cilia in the P14 control and *Mak Ick* DKO photoreceptors observed by electron microscopy. Arrowheads indicate basal bodies. Connecting cilia were absent in the *Mak Ick* DKO retina. **(F)** Schematic representation of retinal photoreceptor cilia in the control and *Mak Ick* DKO mice. Although basal bodies were observed, ciliary axonemes and outer segments were not observed in the *Mak Ick* DKO retina. **(G, H)** ERG analysis of *Mak Ick* DKO mice. **(G)** Representative scotopic and photopic ERGs elicited by four different stimulus intensities (−4.0 to 1.0 log cd s/m² and −0.5 to 1.0 log cd s/m², respectively) from the control, *Mak*−/−, and *Mak Ick* DKO

that Mak activation can rescue the abnormalities caused by *Ick* deficiency and vice versa.

## Ciliary defects resulting from cytoplasmic dynein inhibition are rescued by Mak and Ick activation

Given that deficiency of *Mak* and *Ick* impaired retrograde IFT, we hypothesized that Mak and Ick activation could also restore the ciliary abnormalities caused by retrograde IFT defects related to other gene mutations. To test this hypothesis, NIH3T3 cells were transfected with plasmids expressing Ick and treated with Ciliobrevin D, a cytoplasmic dynein inhibitor, and immunofluorescence analysis was performed to observe cilia. We found that the reduced number of cilia in cells treated with Ciliobrevin D is partially restored by Ick expression (Fig S6H and I). Next, we performed knockdown and rescue experiments to observe cilia. We constructed shRNAs to knock down *Dync2li1*, a ciliopathy gene encoding cytoplasmic dynein-2 light intermediate chain 1, and confirmed that Dync2li1 expression levels decreased in cells expressing Dync2li1-shRNA2, Dync2li1-shRNA3, Dync2li1-shRNA5, and Dync2li1-shRNA6 (Fig S6J). We transfected the Control-shRNA, Dync2li1-shRNA3, or Dync2li1-shRNA5 expression plasmids into NIH3T3 cells with plasmids encoding Ick and examined the cilia by immunostaining using an anti-Actub antibody. We found that the expression of Ick rescued Dync2li1-shRNA-induced ciliary elongation (Figs 4M and N and S6K and L). We also transfected the Control-shRNA or Dync2li1-shRNA3 expression plasmids into NIH3T3 cells, treated the cells with BGJ398, and observed that Dync2li1-shRNA-mediated ciliary elongation was suppressed by BGJ398 treatment (Fig 4O and P). To examine whether aberrant Hedgehog signaling because of dynein inhibition is rescued by Ick and Mak activation, we performed a reporter gene assay using a NanoLuc luciferase reporter construct driven by 8x Gli1-binding sites and a minimal promoter (Fig 4Q). Luciferase activity increased in cells expressing Dync2li1-shRNA2, Dync2li1-shRNA3, and Dync2li1-shRNA6 compared with that in cells expressing Control-shRNA, indicating that Hedgehog signaling was activated by *Dync2li1* knockdown (Figs 4R and S and S6M). We found that the expression of Ick and Mak suppressed *Dync2li1* knockdown–induced Hedgehog signaling activation (Fig 4R and S). These results demonstrate that Ick and Mak activation can rescue ciliary defects caused by cytoplasmic dynein inhibition.

## *Ccrk* depletion causes severe progressive retinal degeneration very similar to that observed in *Mak Ick* DKO mice

To elucidate regulatory mechanisms of Mak and Ick activity in retinal photoreceptor cells, we focused on phosphorylation of the Mak and Ick proteins, because high-throughput phosphoproteomic analyses have shown that Mak and Ick are extensively phosphorylated at residue 157 (PhosphoSitePlus) (60), which is located in the kinase domain showing a high similarity of amino acid sequence and predicted structure between Mak and Ick (Fig 5A). To examine whether this residue is phosphorylated in the retina, we performed a Western blot analysis using an antibody against phosphorylated Mak Thr-157 (pMak) and detected pMak in the $Mak^{+/+}$ retina (Fig 5B). To investigate the physiological role of Thr-157 phosphorylation, we generated a construct encoding the Ick protein harboring a Thr-to-Ala mutation at residue 157 (Ick-T157A). NIH3T3 cells were transfected with plasmids expressing EGFP, wild-type Ick (Ick-WT), Ick-T157A, or Ick harboring a Lys-to-Arg mutation at residue 33 (Ick-kinase dead [KD]), and immunofluorescence analysis was performed using an antibody against phosphorylated Kif3a Thr-674 (pKif3a). As previously observed (33), cytoplasmic pKif3a signals increased in cells expressing Ick-WT compared with those in cells expressing EGFP (Fig 5C). In contrast, the expression of Ick-T157A and Ick-KD did not increase cytoplasmic pKif3a signals, suggesting that Thr-157 phosphorylation is required for Mak and Ick activity (Fig 5C). To identify kinase(s) phosphorylating Thr-157, we searched for the serine–threonine kinases belonging to "Kinase, in vitro" and/or top 5 kinases listed in the "Kinase Prediction" tool (61) for phosphorylation of Mak and Ick Thr-157 using PhosphoSitePlus, and picked up Ccrk, Map3k5, Map3k15, Map2k2, Stk16, Mos, and Stk32b (Fig 5D). Among the genes encoding them, a previous ChIP-seq analysis showed that the binding site of Crx, a master transcription factor in retinal photoreceptor maturation and survival (62, 63), is only assigned to the *Ccrk* gene, suggesting that *Ccrk* can be a candidate gene expressed in retinal photoreceptor cells (Fig 5D) (64). We examined the tissue distribution of the transcripts of *Ccrk* and *Bromi/Tbc1d32*, which encodes a physical and functional interaction partner of Ccrk (65), by RT–PCR analysis using mouse tissue cDNAs and found that *Ccrk* and *Bromi* are ubiquitously expressed in various tissues, including the retina (Fig S7A). UMAP plots of neuronal and glial cells from adult human retinas (57) showed the expression of *CCRK* and *BROMI* in the retinal photoreceptor cells (Figs 3A and 5E). We confirmed that *CCRK* and *BROMI* are expressed in the human retina by RT–PCR analysis (Fig S7B). To examine the expression pattern of *Ccrk* in the retina, we performed an in situ hybridization analysis using mouse retinal sections and observed that *Ccrk* is expressed in the ONL (Fig S7C). Based on these, we focused on Ccrk in subsequent experiments.

To investigate the role of Ccrk in retinal photoreceptor development and maintenance, we generated *Ccrk* flox mice using targeted gene disruption (Fig S7D). We mated *Ccrk* flox mice with *Dkk3-Cre* mice to generate *Ccrk* CKO mice (Fig S7D). In the *Ccrk* CKO retina, no *Ccrk* mRNA or Ccrk protein expression was detected by RT–PCR and Western blotting with the anti-Ccrk antibody that we generated (Fig S7E–G). We first performed histological analyses using retinal sections from *Ccrk* CKO mice at P14 and 1 mo. Toluidine blue staining showed a progressive decrease in the ONL thickness in the *Ccrk* CKO retinas compared with that in the control retinas (Fig 5F and G). Immunohistochemical examination using marker antibodies showed mislocalization of Rhodopsin, S-opsin, and M-opsin in the inner part of the retinal photoreceptor cells in *Ccrk* CKO mice (Figs 5H and S7H). We observed no obvious rod or cone

---

mice at 1 mo. **(H)** Scotopic and photopic amplitudes of a- and b-waves are shown as a function of the stimulus intensity. Data are presented as the mean ± SD. One-way ANOVA followed by Tukey's multiple comparisons test, Control versus $Mak^{-/-}$ indicated by asterisks, *$P < 0.05$, **$P < 0.01$, ***$P < 0.001$, ****$P < 0.0001$, and $Mak^{-/-}$ versus *Mak Ick* DKO indicated by hash symbols, #$P < 0.05$, ##$P < 0.01$, ###$P < 0.001$, ####$P < 0.0001$, n = 4, 4, and 3 mice (control, $Mak^{-/-}$, and *Mak Ick* DKO, respectively). Nuclei were stained with DAPI. OS, outer segment; IS, inner segment; ONL, outer nuclear layer; INL, inner nuclear layer; GCL, ganglion cell layer.

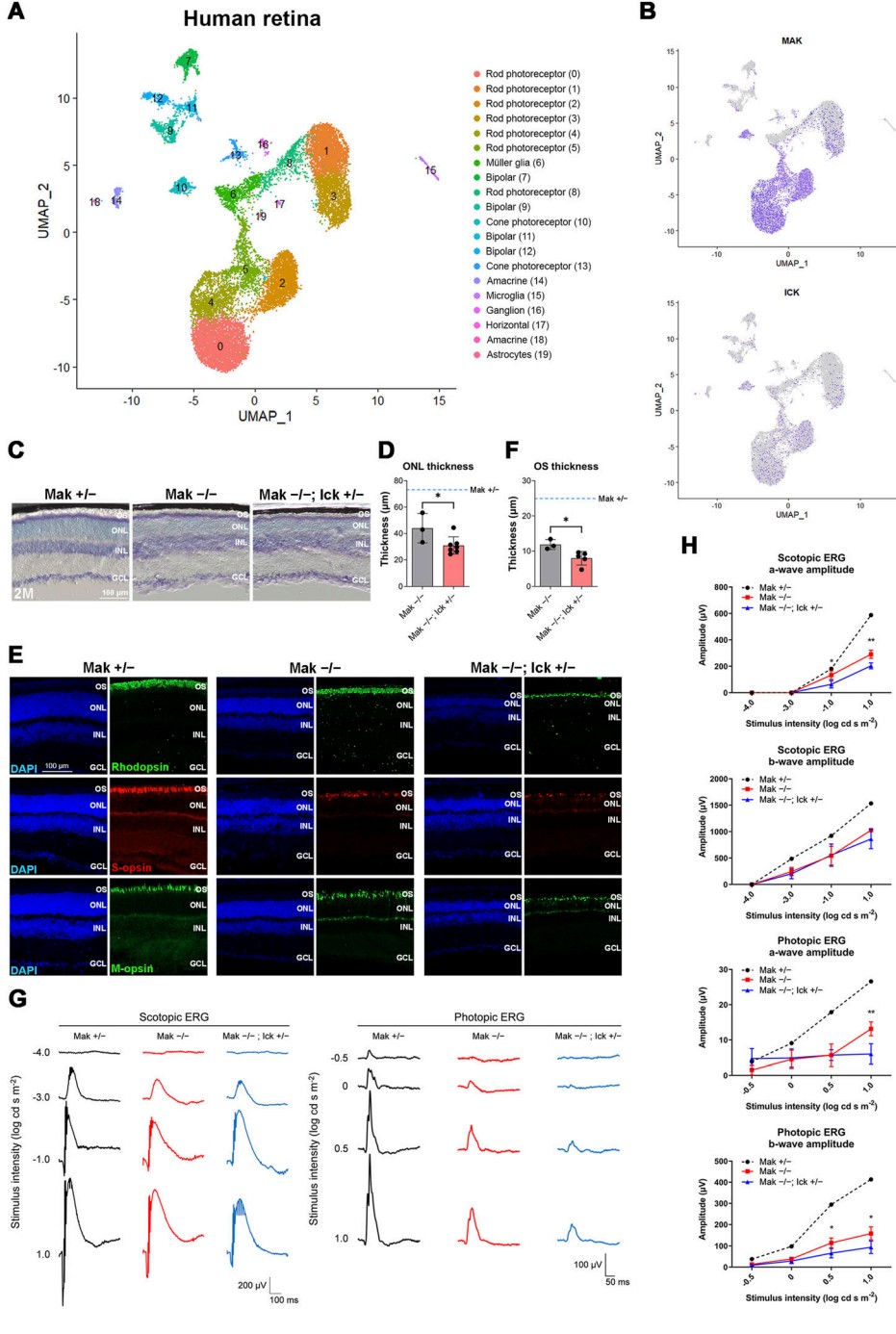

**Figure 3. _Ick_ haploinsufficiency exacerbates retinal degeneration caused by loss of _Mak_.**

**(A)** Uniform Manifold Approximation and Projection visualization of retinal cells in adult human retinas. **(B)** Feature plots showing the expression of _MAK_ and _ICK_ in retinal cells of adult human retinas. **(C, D)** Toluidine blue staining of retinal sections from $Mak^{+/-}$, $Mak^{-/-}$, and $Mak^{-/-}$; $Ick^{+/-}$ mice at 2 mo. The ONL thickness was measured. Data are presented as the mean ± SD. *$P < 0.05$ (unpaired $t$ test). n = 5, 3, and 7 mice ($Mak^{+/-}$, $Mak^{-/-}$, and $Mak^{-/-}$; $Ick^{+/-}$, respectively). The blue dotted line indicates the ONL thickness in the $Mak^{+/-}$ retina. **(E, F)** Immunostaining of retinal sections from $Mak^{+/-}$, $Mak^{-/-}$, and $Mak^{-/-}$; $Ick^{+/-}$ mice at 2 mo using marker antibodies against Rhodopsin, S-opsin, and M-opsin. Nuclei were stained with DAPI. The rod outer segment length was measured. Data are presented as the mean ± SD. *$P < 0.05$ (unpaired $t$ test). n = 4, 3, and 5 mice ($Mak^{+/-}$, $Mak^{-/-}$, and $Mak^{-/-}$; $Ick^{+/-}$, respectively). The blue dotted line indicates the rod outer segment length in the $Mak^{+/-}$ retina. **(G, H)** ERG analysis of $Mak^{-/-}$; $Ick^{+/-}$ mice at 2 mo. **(G)** Representative scotopic and photopic ERGs elicited by four different stimulus intensities (−4.0 to 1.0 log cd s/m² and −0.5 to 1.0 log cd s/m², respectively) from $Mak^{+/-}$, $Mak^{-/-}$, and $Mak^{-/-}$; $Ick^{+/-}$ mice. **(H)** Scotopic and photopic amplitudes of a- and b-waves are shown as a function of the stimulus intensity. Data are presented as the mean ± SD. *$P < 0.05$, **$P < 0.01$ (unpaired $t$ test). n = 5, 3, and 6 mice ($Mak^{+/-}$, $Mak^{-/-}$, and $Mak^{-/-}$; $Ick^{+/-}$, respectively). OS, outer segment; ONL, outer nuclear layer; INL, inner nuclear layer; GCL, ganglion cell layer.

outer segment structures in _Ccrk_ CKO mice (Figs 5H and S7H). To examine whether _Ccrk_ deficiency affects ciliary formation and the distribution of IFT components in the cilia of retinal photoreceptor cells, we immunostained retinal sections from _Ccrk_ CKO mice at P14 using antibodies against Actub, Cep164 (a basal body marker), IFT88, and Mak. Similar to Ick, Mak was localized in the distal regions of the photoreceptor ciliary axonemes in the control retina (Fig 5I). In contrast, photoreceptor ciliary axonemes were not observed in the _Ccrk_ CKO retinas (Fig 5I and J). We observed that IFT88 signals were concentrated near the basal bodies of retinal photoreceptor cells

in _Ccrk_ CKO mice (Fig 5I). To evaluate the electrophysiological properties of the _Ccrk_ CKO retina, we performed an ERG analysis and observed no significant ERG responses in _Ccrk_ CKO mice (Fig 5K). The observed phenotypes of severe retinal degeneration in _Ccrk_ CKO mice resemble those observed in _Mak Ick_ DKO mice, suggesting that the Ccrk-Mak/Ick axis plays a crucial role in IFT regulation in retinal photoreceptor cells. To confirm this, we performed a Western blot analysis and found that pMak was markedly decreased in the _Ccrk_ CKO retina compared with that in the control retina (Fig 5L).

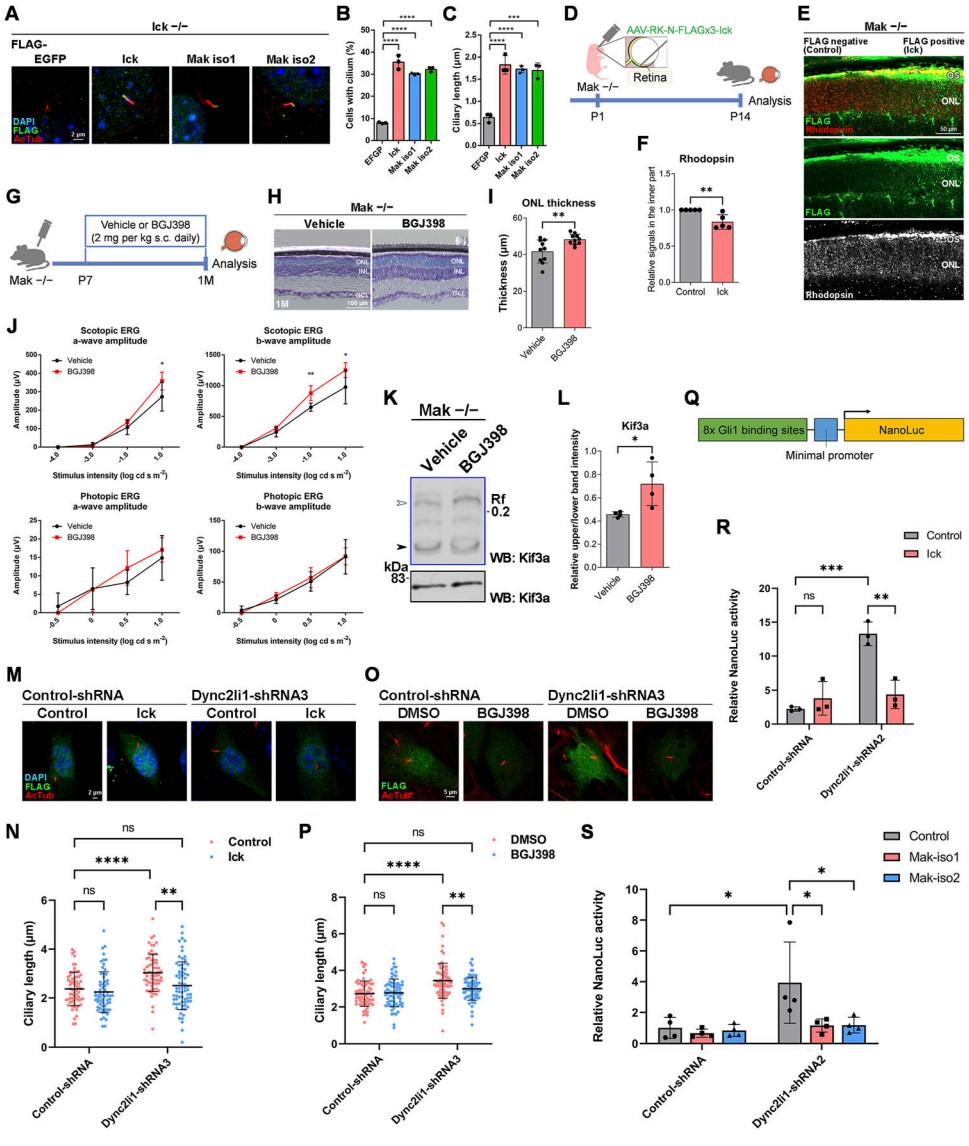

**Figure 4. Activation of Mak and Ick can rescue the abnormalities caused by *Ick*, *Mak*, and *Dync2li1* deficiency.**

**(A, B, C)** Effects of Mak overexpression on ciliary defects in *Ick⁻/⁻* MEFs. **(A)** Plasmid encoding a FLAG-tagged EGFP, Ick, Mak isoform1 (Mak iso1), or Mak isoform2 (Mak iso2) was transfected into *Ick⁻/⁻* MEFs. Cells were immunostained with anti-FLAG and anti-Actub antibodies. **(B, C)** Numbers (B) and length (C) of the cilia stained with an antibody against Actub in FLAG-positive cells were measured. Data are presented as the mean ± SD. ***$P < 0.001$, ****$P < 0.0001$ (one-way ANOVA followed by Tukey's multiple comparisons test). n = 3 experiments. **(D, E, F)** Subcellular localization of Rhodopsin in Ick-overexpressing photoreceptor cells of the *Mak⁻/⁻* mouse retina. **(D)** Schematic diagram of schedule for AAV subretinal injection and harvest of retinas. An AAV expressing FLAG-tagged Ick driven by the Rhodopsin kinase promoter was injected into P1 *Mak⁻/⁻* mouse retinas. At P14, their retinas were harvested, sectioned, and immunostained with anti-FLAG and anti-Rhodopsin antibodies. **(E)** Rhodopsin signals in the inner part of photoreceptors decreased in FLAG-positive regions. **(F)** Immunofluorescence signals of Rhodopsin detected in the inner part of photoreceptors were quantified using ImageJ software. The signals of Rhodopsin in the inner part of photoreceptors were normalized to the total (OS + the inner part) signals of Rhodopsin in photoreceptors. Rhodopsin signals in the inner part (normalized to the total signals) of FLAG-positive regions relative to those of FLAG-negative regions were then calculated. Data are presented as the mean ± SD. **$P < 0.01$ (unpaired *t* test). n = 5 retinas from four mice. **(G)** Schematic diagram of schedule for drug administration. BGJ398, an inhibitor of Fgfrs, was injected into *Mak⁻/⁻* mice from P7 to 1 mo every day. s.c., subcutaneous. **(H, I)** Toluidine blue staining of retinal sections from 1 mo *Mak⁻/⁻* mice treated with or without BGJ398, which was injected into the mice from P7 to 1 mo every day. Data are presented as the

mean ± SD. **$P < 0.01$ (unpaired *t* test). n = 10 mice each. **(J)** ERG analysis of 1 mo *Mak⁻/⁻* mice treated with or without BGJ398, which was injected into the mice from P7 to 1 mo every day. The scotopic and photopic amplitudes of a- and b-waves are shown as a function of the stimulus intensity (−4.0 to 1.0 log cd s/m² and −0.5 to 1.0 log cd s/m², respectively). Data are presented as the mean ± SD. *$P < 0.05$, **$P < 0.01$ (unpaired *t* test). n = 6 mice each (scotopic), n = 5 and 6 mice (photopic) (vehicle-treated and BGJ398-treated, respectively). **(K, L)** Kif3a phosphorylation in the retina of 1 mo *Mak⁻/⁻* mice treated with or without BGJ398, which was injected into the mice from P7 to 1 mo every day. **(K)** Retinal lysates were analyzed by Phos-tag (blue box) and conventional (black box) Western blotting with the anti-Kif3a antibody. **(L)** Relative phosphorylation of Kif3a was quantified by the ratio of the upper band intensity (white arrowhead) to the lower band intensity (black arrowhead). Data are presented as the mean ± SD. *$P < 0.05$ (unpaired *t* test). n = 4 mice each. **(M, N)** Effects of Ick overexpression on ciliary length in cells knocked down for *Dync2li1*. **(M)** Plasmid encoding Control-shRNA or Dync2li1-shRNA3 was co-transfected into NIH3T3 cells with or without a plasmid expressing Ick in combination with a construct encoding FLAG-tagged EGFP. Cells were immunostained with anti-FLAG and anti-Actub antibodies. **(N)** Length of cilia stained with an antibody against Actub in FLAG-positive cells was measured. Data are presented as the mean ± SD. **$P < 0.01$, ****$P < 0.0001$, ns, not significant (two-way ANOVA followed by Tukey's multiple comparisons test). Control-shRNA; Control, Control-shRNA; Ick, Dync2li1-shRNA3; Control, and Dync2li1-shRNA3; Ick, n = 66, 63, 67, and 70 cilia, respectively, from three experiments. **(O, P)** Effects of BGJ398 treatment on ciliary length in cells knocked down for *Dync2li1*. **(O)** NIH3T3 cells transfected with plasmids expressing FLAG-tagged EGFP and Control-shRNA or Dync2li1-shRNA3 were treated with 100 nM BGJ398 or DMSO for 24 h before harvest. Cells were immunostained with anti-FLAG and anti-Actub antibodies. **(P)** Length of cilia stained with an antibody against Actub in FLAG-positive cells was measured. Data are presented as the mean ± SD. **$P < 0.01$, ****$P < 0.0001$, ns, not significant (two-way ANOVA followed by Tukey's multiple comparisons test). Control-shRNA; DMSO, Control-shRNA; BGJ398, Dync2li1-shRNA3; DMSO, and Dync2li1-shRNA3; BGJ398, n = 71, 70, 74, and 73 cilia, respectively, from three experiments. **(Q, R, S)** Luciferase reporter gene assay using 8x Gli1-binding sites–minimal promoter–NanoLuc luciferase constructs. **(Q)** Schematic diagram of the construct expressing NanoLuc luciferase under the control of 8x Gli1-binding sites and minimal promoter. **(R, S)** NIH3T3 cells were transfected with plasmids expressing Dync2li1-shRNA2 and Ick (R), Mak iso1, or Mak iso2 (S) along with a NanoLuc luciferase reporter construct driven by the 8x Gli1-binding sites and minimal promoter and a Firefly luciferase–expressing construct driven by the SV40 promoter and enhancer. Luciferase activities of cell lysates were measured 24 h after serum starvation with 100 nM smoothened agonist (SAG). NanoLuc luciferase activity was normalized to Firefly luciferase activity. Data are presented as the mean ± SD. *$P < 0.05$, **$P < 0.01$, ***$P < 0.001$, ns, not significant (two-way ANOVA followed by Tukey's multiple comparisons test). n = 3 (R) and n = 4 (S) experiments. Nuclei were stained with DAPI. OS, outer segment; ONL, outer nuclear layer; INL, inner nuclear layer; GCL, ganglion cell layer.

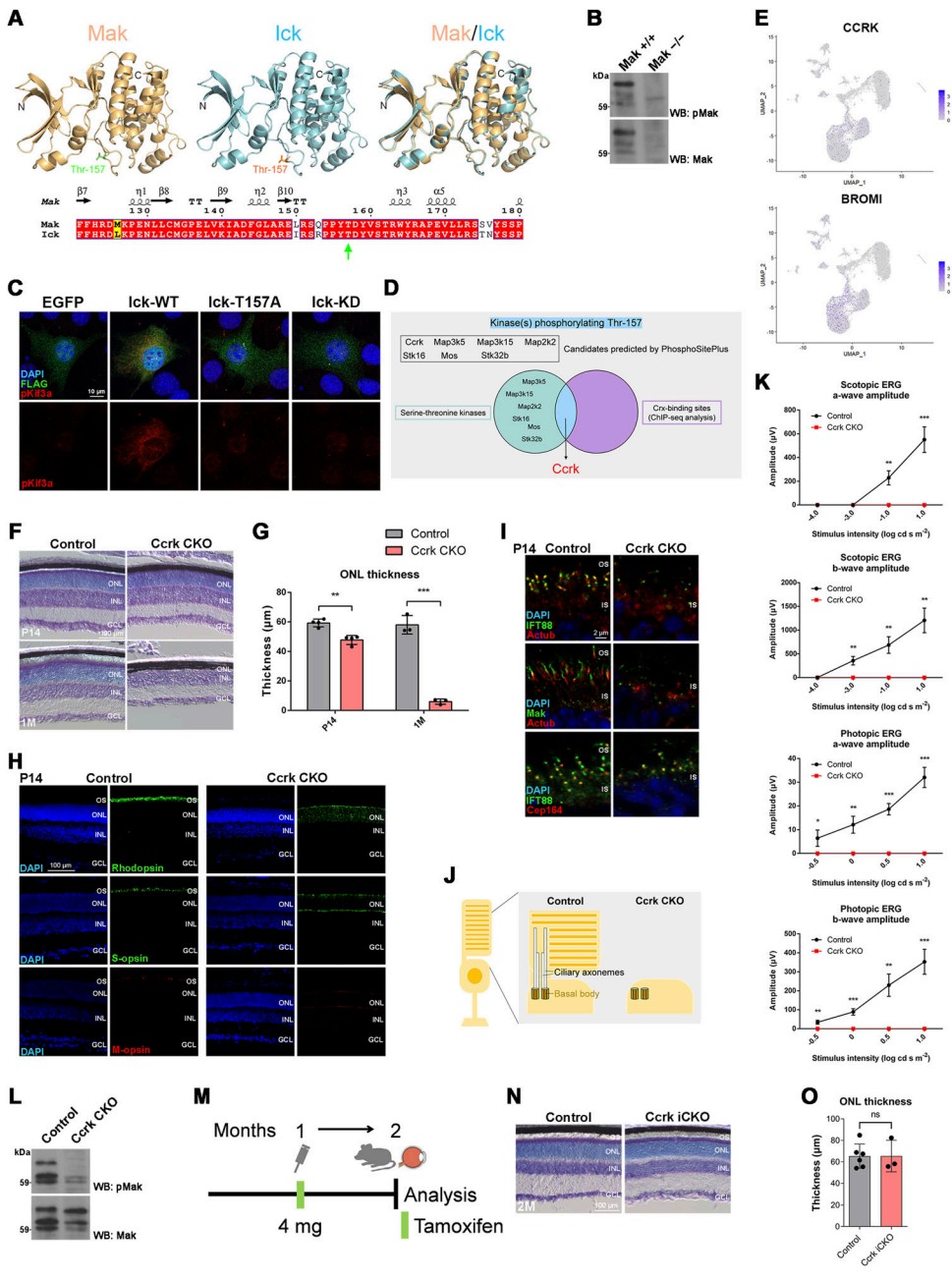

**Figure 5. Severe progressive retinal degeneration in *Ccrk* CKO mice.**
**(A)** Thr-157 in the Mak and Ick proteins. (Upper panel) Predicted 3D structures of the kinase domains of the Mak and Ick proteins and a superposition of their structures are shown. (Lower panel) Structure-based sequence alignment of the Mak and Ick proteins prepared with ESPript. Similar and identical residues are marked by yellow and red boxes, respectively. The secondary structure assignment is based on the predicted structure of Mak. A green arrow indicates Thr-157. **(B)** Western blot analysis of the phosphorylated Mak Thr-157 (pMak) and Mak protein in *Mak*⁺/⁺ and *Mak*⁻/⁻ retinas. **(C)** Kif3a phosphorylation in Ick-overexpressing cells. A plasmid encoding a FLAG-tagged EGFP, wild-type Ick (Ick-WT), Ick harboring a Thr-to-Ala mutation at residue 157 (Ick-T157A), or Ick harboring a Lys-to-Arg mutation at residue 33 (Ick-kinase dead [KD]) was transfected into NIH3T3 cells. Cells were immunostained with anti-FLAG and anti-phosphorylated Kif3a Thr-674 (pKif3a) antibodies. Substantial cytoplasmic pKif3a signals were observed in cells expressing Ick-WT but not in those expressing EGFP, Ick-T157A, or Ick-KD. **(D)** Schematic diagram summarizing search for kinase(s) phosphorylating Mak and Ick Thr-157. **(E)** Feature plots showing the expression of *CCRK* and *BROMI* in retinal cells of adult human retinas. **(F, G)** Toluidine blue staining of retinal sections from the control and *Ccrk* CKO mice at P14 and 1 mo. The ONL thickness was measured. Data are presented as the mean ± SD. **P < 0.01, ***P < 0.001 (unpaired *t* test). n = 4 mice per each genotype at P14 and n = 3 mice per each genotype at 1 mo. **(H)** Immunostaining of retinal sections from the control and *Ccrk* CKO mice at P14 using marker antibodies against Rhodopsin, S-opsin, and M-opsin. Severe photoreceptor outer segment disorganization and mislocalization of Rhodopsin and cone opsins were observed in the *Ccrk* CKO retina. **(I)** Ciliary localization of IFT components in photoreceptor cells of the *Ccrk* CKO retina. Retinal sections obtained from the control and *Ccrk* CKO mice at P14 were immunostained using antibodies against IFT88, Actub, Mak, and Cep164 (a marker for basal bodies). The ciliary axoneme of retinal photoreceptor cells was absent in *Ccrk* CKO mice. **(J)** Schematic representation of retinal photoreceptor cilia in the control and *Ccrk* CKO mice. Although basal bodies were observed, ciliary axonemes and outer segments were not observed in the *Ccrk* CKO retina. **(K)** ERG analysis of *Ccrk* CKO mice. Scotopic and photopic ERGs elicited by four different stimulus intensities (−4.0 to 1.0 log cd s/m² and −0.5 to 1.0 log cd s/m², respectively) from the control and *Ccrk* CKO mice at 1 mo. The scotopic and photopic amplitudes of a- and b-waves are shown as a function of the stimulus intensity. Data are presented as the mean ± SD. *P < 0.05, **P < 0.01, ***P < 0.001 (unpaired *t* test). n = 3 mice per each genotype. **(L)** Western blot analysis of pMak and the Mak protein in the control and *Ccrk* CKO retinas. **(M)** Schematic diagram of schedule for tamoxifen administration and analysis of mice. Mice were injected with tamoxifen at 1 mo and analyzed at 2 mo. **(N, O)** Toluidine blue staining of retinal sections from the control and *Ccrk* iCKO mice at 2 mo. The ONL thickness was measured. Data are presented as the mean ± SD. ns, not significant (unpaired *t* test). n = 6 and 3 mice (control and *Ccrk* iCKO, respectively). Nuclei were stained with DAPI. OS, outer segment; IS, inner segment; ONL, outer nuclear layer; INL, inner nuclear layer; GCL, ganglion cell layer.

## Ccrk is dispensable for Mak phosphorylation maintenance in vivo

To investigate whether Ccrk is required for pMak maintenance in vivo, we mated *Ccrk* flox mice with *Crx-CreERT2* mice (66, 67), which express *CreERT2* predominantly in retinal photoreceptor cells, injected tamoxifen into *Ccrk*^flox/flox^; *Crx-CreERT2* mice at

1 mo, and analyzed the mice at 2 mo (Fig 5M). RT–PCR and Western blotting showed that *Ccrk* mRNA and Ccrk protein expression decreased in the retina of *Ccrk*^flox/flox^; *Crx-CreERT2* mice treated with tamoxifen (*Ccrk* iCKO mice) (Fig S7I and J). Toluidine blue staining showed no significant differences in the ONL thickness between the control and *Ccrk* iCKO retinas (Fig 5N and O).

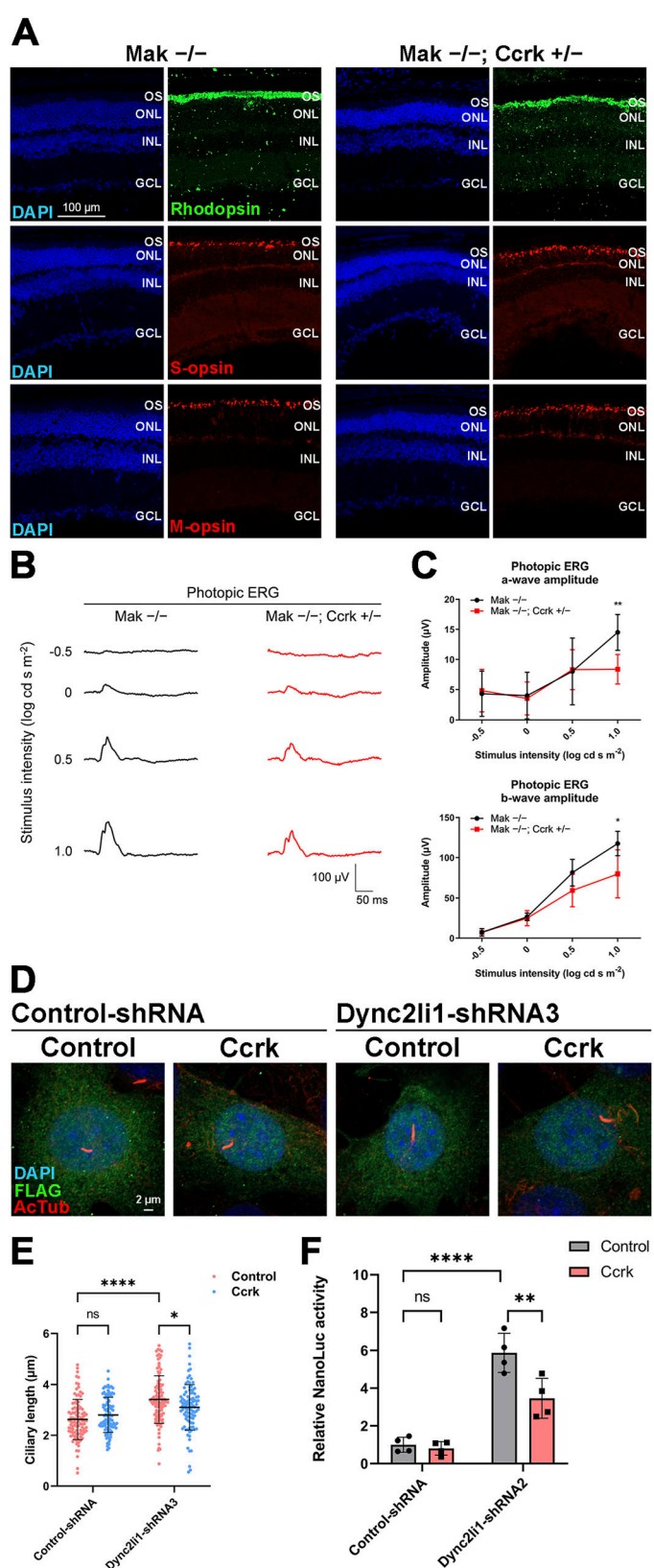

**Figure 6. Effects of *Ccrk* dosage on retinal degeneration in *Mak*<sup>−/−</sup> mice and ciliary abnormalities caused by *Dync2li1* deficiency.**
(A) Immunostaining of retinal sections from *Mak*<sup>−/−</sup> and *Mak*<sup>−/−</sup>; *Ccrk*<sup>+/−</sup> mice at 2 mo using marker antibodies against Rhodopsin, S-opsin, and M-opsin. Rod outer segment disorganization and mislocalization of cone opsins were enhanced in the *Mak*<sup>−/−</sup>; *Ccrk*<sup>+/−</sup> retina. OS, outer segment; ONL, outer nuclear layer; INL, inner nuclear

Immunohistochemical examination using marker antibodies against Rhodopsin, S-opsin, and M-opsin also showed no obvious differences between the control and *Ccrk* iCKO retinas (Fig S7K). In addition, we performed an ERG analysis and found no significant differences in the amplitudes of scotopic and photopic ERG a- and b-waves between the control and *Ccrk* iCKO mice (Fig S7L and M). Furthermore, we found no substantial differences in the amount of pMak between the control and *Ccrk* iCKO retinas (Fig S7N). These results suggest that pMak is maintained by autophosphorylation (68) but not by Ccrk in vivo.

### *Ccrk* dosage affects retinal degeneration in *Mak*$^{-/-}$ mice and ciliary abnormalities caused by cytoplasmic dynein inhibition

To assess the relationship between Ccrk and Mak/Ick in retinal photoreceptor cells, we sought to generate and analyze *Mak*$^{-/-}$; *Ccrk*$^{+/-}$ mice. To examine the effects of *Ccrk* heterozygous disruption on retinal photoreceptor cells, we compared the retinal phenotypes of *Mak*$^{+/-}$; *Ccrk*$^{+/-}$ mice with those of *Mak*$^{+/-}$ mice at 2 mo. Immunohistochemical examination using marker antibodies against Rhodopsin, S-opsin, and M-opsin showed no obvious differences between the *Mak*$^{+/-}$ and *Mak*$^{+/-}$; *Ccrk*$^{+/-}$ retinas (Fig S8A). To evaluate the effects of *Ccrk* heterozygous deficiency on retinal function, we performed an ERG analysis and found no significant differences in the amplitudes of scotopic a- and b-waves and photopic b-waves between *Mak*$^{+/-}$ and *Mak*$^{+/-}$; *Ccrk*$^{+/-}$ mice, although the photopic a-wave amplitude significantly but slightly decreased in *Mak*$^{+/-}$; *Ccrk*$^{+/-}$ mice compared with that in *Mak*$^{+/-}$ mice (Fig S8B and C). These results show that *Ccrk* heterozygous deficiency alone hardly affects retinal photoreceptor function and maintenance.

Next, we performed histological analyses of retinal sections from *Mak*$^{-/-}$; *Ccrk*$^{+/-}$ mice at 2 mo. Immunohistochemical examination showed more disorganized structures of rod outer segments in *Mak*$^{-/-}$; *Ccrk*$^{+/-}$ mice than in *Mak*$^{-/-}$ mice (Fig 6A). Mislocalization of cone opsins to the inner part of photoreceptors was more prominent in the *Mak*$^{-/-}$; *Ccrk*$^{+/-}$ retina than in the *Mak*$^{-/-}$ retina (Fig 6A). To evaluate the electrophysiological properties of the *Mak*$^{-/-}$; *Ccrk*$^{+/-}$ retina, we performed an ERG analysis and found that photopic a- and b-wave amplitudes in *Mak*$^{-/-}$; *Ccrk*$^{+/-}$ mice were significantly lower than those in *Mak*$^{-/-}$ mice (Figs 6B and C and S8D and E). Although we observed that the photopic a-wave amplitudes to the brightest stimulus were decreased by *Ccrk* heterozygous deficiency in both *Mak*$^{+/-}$ and *Mak*$^{-/-}$ mice, the reduction rate was higher in

*Mak*$^{-/-}$ mice (42%) than in *Mak*$^{+/-}$ mice (19%) (Figs 6B and C and S8B and C). These results suggest that *Ccrk* is a modifier of retinal degeneration observed in *Mak*$^{-/-}$ mice and support the idea that the Ccrk-Mak/Ick axis functions in retinal photoreceptor cells.

To test whether Ccrk activation can restore the ciliary abnormalities caused by cytoplasmic dynein defects, we transfected the Control-shRNA or Dync2li1-shRNA3 expression plasmids into NIH3T3 cells with plasmids encoding Ccrk and examined the cilia by immunostaining using the anti-Actub antibody. We found that Ccrk expression rescued Dync2li1-shRNA-induced ciliary elongation (Fig 6D and E). We also observed that Ccrk expression suppresses *Dync2li1* knockdown–induced Hedgehog signaling activation (Fig 6F). These results suggest that activation of the Ccrk-Mak/Ick axis can rescue ciliary defects caused by cytoplasmic dynein inhibition.

## Discussion

In the current study, we performed molecular, cellular, histological, and electrophysiological analyses in combination with mouse genetics and identified Ccrk-Mak/Ick kinase signaling as an IFT regulator essential for retinal photoreceptor maintenance. In addition, our results shed light on pathological mechanisms underlying retinitis pigmentosa caused by mutations in the human *MAK* gene and suggest that activation of Ick is a potential therapeutic approach to treat this disease. We previously reported that Ick regulates the IFT turnaround step at the tip of the cilia (33). IFT-A, IFT-B, and BBSome components are accumulated at the ciliary tips in *Ick*$^{-/-}$ MEFs (33). Although Ick is recognized as a critical regulator of the IFT turnaround process (16, 27), there were no obvious differences in the distribution of IFT components in the photoreceptor ciliary axonemes between the control and *Ick* CKO retinas. This observation prompted us to screen other serine–threonine kinase(s), and Mak was identified as a candidate regulator of the IFT turnaround step. We observed that Mak and Ick localized to the ciliary tips in cultured cells and to the distal region of photoreceptor ciliary axonemes in the retina. It was previously reported that Ick localization to the ciliary tip is mediated through anterograde trafficking by IFT-B (44). A recent study showed a critical role of IFT-A·Tubby-like protein carriages in the ciliary tip localization of Ick (16). Using these mechanisms, Mak and Ick may be transported to the distal region of the ciliary axonemes in the retinal photoreceptor cells. We also observed that the overexpression of Mak and Ick similarly changed the ciliary localization of IFT components

---

layer; GCL, ganglion cell layer. **(B, C)** ERG analysis of *Mak*$^{-/-}$; *Ccrk*$^{+/-}$ mice at 2 mo. **(B)** Representative photopic ERGs elicited by four different stimulus intensities (−0.5 to 1.0 log cd s/m²) from *Mak*$^{-/-}$ and *Mak*$^{-/-}$; *Ccrk*$^{+/-}$ mice. **(C)** Photopic amplitudes of a- and b-waves are shown as a function of the stimulus intensity. Data are presented as the mean ± SD. *$P < 0.05$, **$P < 0.01$ (unpaired *t* test). n = 7 and 4 mice (*Mak*$^{-/-}$ and *Mak*$^{-/-}$; *Ccrk*$^{+/-}$, respectively). **(D, E)** Effects of Ccrk overexpression on ciliary length in cells knocked down for *Dync2li1*. **(D)** Plasmid encoding Control-shRNA or Dync2li1-shRNA3 was co-transfected into NIH3T3 cells with or without a plasmid expressing Ccrk in combination with a construct encoding FLAG-tagged EGFP. Cells were immunostained with anti-FLAG and anti-Actub antibodies. **(E)** Length of cilia stained with an antibody against Actub in FLAG-positive cells was measured. Data are presented as the mean ± SD. *$P < 0.05$, ****$P < 0.0001$, ns, not significant (two-way ANOVA followed by Tukey's multiple comparisons test). Control-shRNA; Control, Control-shRNA; Ccrk, Dync2li1-shRNA3; Control, and Dync2li1-shRNA3; Ccrk, n = 100, 104, 108, and 101 cilia, respectively, from four experiments. **(F)** Luciferase reporter gene assay using 8x Gli1-binding sites–minimal promoter–NanoLuc luciferase constructs. NIH3T3 cells were transfected with plasmids expressing Dync2li1-shRNA2 and Ccrk along with a NanoLuc luciferase reporter construct driven by the 8x Gli1-binding sites and minimal promoter and a Firefly luciferase–expressing construct driven by the SV40 promoter and enhancer. Luciferase activities of cell lysates were measured 24 h after serum starvation with 100 nM SAG. NanoLuc luciferase activity was normalized to Firefly luciferase activity. Data are presented as the mean ± SD. **$P < 0.01$, ****$P < 0.0001$, ns, not significant (two-way ANOVA followed by Tukey's multiple comparisons test). N = 4 experiments.

in cultured cells. IFT components were concentrated at the tips of the photoreceptor ciliary axonemes in the $Mak^{-/-}$ retina, which is similar to observations in $Ick^{-/-}$ MEFs (33). Phosphorylation of Kif3a, an Ick substrate (33), was lower in the $Mak^{-/-}$ retina than in the $Mak^{+/+}$ retina. In addition, the deletion of both $Mak$ and $Ick$ leads to severe retinal degeneration. Based on these findings, we concluded that Mak and Ick play a central role in IFT turnaround regulation in retinal photoreceptor cells, although we cannot exclude the possibility that kinase(s) other than Ick and Mak function as regulators of the IFT turnaround process and that Mak and Ick may have functions other than the regulation of the IFT turnaround process. We previously observed that both $Mak$ and $Ick$ are expressed in the brain, lung, spleen, ovary, and testis, in addition to the retina (69), suggesting that $Mak$ and $Ick$ genetically interact and play a role in IFT regulation not only in the retina but also in other tissues. Non-syndromic retinitis pigmentosa, but not syndromic ciliopathies, exhibited by individuals with $MAK$ gene mutations (54, 55) may be due to genetic interactions between $MAK$ and $ICK$.

We found that Ick Thr-157 phosphorylation is critical for its kinase activity, which is consistent with previous studies indicating its importance in Mak and Ick activity (68, 70, 71). The level of Thr-157 phosphorylation in the Mak protein was markedly decreased in the $Ccrk$ CKO retina compared with that in the control retina. $Ccrk$ CKO mice exhibited progressive retinal degeneration phenotypes, quite similar to those observed in $Mak$ $Ick$ DKO mice. In addition, $Mak^{-/-}$; $Ccrk^{+/-}$ retinas were more severely degenerated than $Mak^{-/-}$ retinas. These observations strongly suggest that Ccrk acts as a major upstream regulator of Mak and Ick in retinal photoreceptor cells, which is supported by previous reports showing that Ccrk directly phosphorylates Ick and Mak at Thr-157 in vitro (68, 71). However, we cannot exclude the possibility that Mak and Ick activity is regulated by other mechanism(s). It was previously reported that protein phosphatase 5 (PP5) dephosphorylates Ick at Thr-157 (71), implying that Mak and Ick are negatively regulated by PP5 in retinal photoreceptor cells. Fgfrs interact with, phosphorylate, and inactivate Ick (59) and are expressed in retinal photoreceptor cells (72, 73, 74, 75, 76, 77, 78). We observed that retinal degeneration in $Mak^{-/-}$ mice was suppressed by BGJ398, an FDA-approved inhibitor of Fgfrs, whereas that in $Mak$ $Ick$ DKO mice could not be rescued by BGJ398. This observation suggests that the protective role of BGJ398 on the $Mak^{-/-}$ retina is mediated through Ick activation, although we cannot exclude the possibility that other pathway(s) are also involved in the effects of BGJ398 treatment. The increased phosphorylation level of Kif3a in the retina of $Mak^{-/-}$ mice treated with BGJ398 supports the idea that Ick is activated in the $Mak^{-/-}$ retina by pharmacological inhibition of Fgfrs. These observations suggest that fibroblast growth factor–Fgfr signaling negatively regulates Ick activity in retinal photoreceptor cells, although Ick activation in retinal cells other than photoreceptors may have contributed to the suppression of retinal degeneration in BGJ398-treated $Mak^{-/-}$ mice.

Photoreceptor ciliary axonemes in $Mak^{-/-}$ retinas were elongated, whereas those in $Mak$ $Ick$ DKO retinas were not observed. Accumulation of IFT components at ciliary tips in cultured cells and cochlear hair cells lacking $Ick$ (33, 43, 44), as well as photoreceptor cells of the $Mak^{-/-}$ retina, shows that retrograde IFT can be inhibited by disruption of $Ick$ and $Mak$. Dync2li1 knockdown and knockout caused elongation and severe shortening of cilia in RPE1 cells, respectively (79, 80), suggesting that ciliary length is affected by the

extent to which retrograde IFT is inhibited. In contrast to the absence or shortening of cilia caused by the loss of $Dync2li1$, a small amount of the Dync2li1 protein remaining in the knockdown cells may induce ciliary elongation. This idea is supported by ciliary elongation in cultured cells knocked down for $Dync2li1$ in our study, and the absence of cilia or ciliary shortening in the ventral node of mouse embryos lacking $Dync2li1$ in a previous study (81). Therefore, the differences in photoreceptor ciliary axonemes between the $Mak^{-/-}$ and $Mak$ $Ick$ DKO retinas may be attributed to the severity of retrograde IFT defects. Ick may compensate for the impairment of retrograde IFT in the $Mak^{-/-}$ retina. We and others have previously reported that loss of function of $Ick$ results in the elongation or shortening/absence of cilia (33, 41, 42, 43, 44, 47, 48, 59), raising the possibility that the total remaining amount and/or activity of Ick and Mak in cells with elongated cilia is higher than that in cells showing the shortening or absence of cilia. Previous studies have shown that IFT components accumulate at the ciliary tips in $Ccrk^{-/-}$ MEFs and $Ccrk$-deficient RPE1 cells (82, 83), supporting the idea that the Ccrk-Mak/Ick signaling pathway regulates the IFT turnaround process in retinal photoreceptor cells. Notably, cilia were elongated in $Ccrk^{-/-}$ MEFs and $Ccrk$-deficient RPE1 cells (82, 83), whereas photoreceptor ciliary axonemes were not observed in the $Ccrk$ CKO retina. A possible explanation for this disparity is the differences between cultured cells and in vivo conditions or the severity of retrograde IFT defects. Dysfunction of retrograde IFT in $Ccrk^{-/-}$ MEFs and $Ccrk$-deficient RPE1 cells may be compensated in cultured conditions. Loss of function of $Ccrk$ orthologues, $lf2$ in $Chlamydomonas$ and $dyf-18$ in $C. elegans$, also causes flagellar/ciliary elongation (38, 39, 84, 85). Thus, in this case, it is unclear whether the data obtained from invertebrates and cultured cell systems can directly apply to mammals in vivo.

How do Mak and Ick regulate the IFT turnaround process at ciliary tips through phosphorylation in retinal photoreceptor cells? We have previously reported that Ick phosphorylates the C-terminal portion of Kif3a, including Thr-674 (33). We observed that Kif3a phosphorylation markedly decreased in the $Mak^{-/-}$ retina compared with that in the $Mak^{+/+}$ retina, suggesting that Mak phosphorylates Kif3a in retinal photoreceptor cells. Phosphorylation of serine–threonine residues, including Thr-674 in the Kif3a C-terminal region, is required for ciliary formation in cultured cells and zebrafish (33). In contrast, MEFs harboring a Thr-to-Ala mutation at residue 674 on Kif3a exhibited slight ciliary elongation without affecting the ciliary localization of IFT88 (86). In $Chlamydomonas$, inhibition of phosphorylation of the kinesin-2 motor subunit FLA8, an orthologue of Kif3b, at Ser-663 led to defects in IFT turnaround at the flagellar tip (87). FLA8 Ser-663 is located in a consensus amino acid sequence for phosphorylation by Ick (71), which is evolutionarily conserved among species, implying that Kif3b phosphorylation by Mak and Ick modulates IFT turnaround at the ciliary tip in retinal photoreceptor cells. $C. elegans$ DYF-5 reduced the tubulin-binding affinity of the IFT-B components IFT-74/81 by phosphorylating IFT74, proposing a model in which DYF-5–mediated phosphorylation of IFT74 promotes tubulin unloading at the ciliary tip (88, 89, 90). Future studies are needed to uncover the downstream mechanisms underlying the regulation of IFT by Mak and Ick in retinal photoreceptor cells.

Some of retinitis pigmentosa patients with the same mutation in the *MAK* gene represent different severities of disease ([91](ref)). We found that $Mak^{-/-}; Ick^{+/-}$ mice exhibited severe retinal degeneration compared with $Mak^{-/-}$ mice and that *MAK* and *ICK* are expressed in photoreceptor cells of the human retina. Heterozygous loss-of-function mutations/variants in the human *ICK* gene ([46](ref), [47](ref), [48](ref), [49](ref)) may cause more severe symptoms in retinitis pigmentosa patients having *MAK* mutations. In addition, mutations or variants in other genes involved in retrograde IFT may contribute to the severity of retinal degeneration caused by *MAK* mutations. Indeed, we observed that $Mak^{-/-}; Ccrk^{+/-}$ retinas were more severely degenerated than $Mak^{-/-}$ retinas. Ccrk physically and functionally interacts with Bromi ([65](ref)). We found that both *CCRK* and *BROMI* were expressed in photoreceptor cells of the human retina. Notably, mutations in the human *BROMI* gene are associated with progressive retinal dystrophy ([92](ref)). Our results may advance our understanding of the pathological mechanisms underlying progressive retinal dystrophy in patients with mutations in the human *BROMI* gene.

We found that Mak can rescue ciliary abnormalities caused by *Ick* deficiency and vice versa. In addition, BGJ398 and the overexpression of Ick, Mak, and Ccrk could rescue ciliary defects because of the deficiency of *Dync2li1*, a ciliopathy gene encoding a subunit of cytoplasmic dynein 2 ([80](ref)), which supports the idea that the Ccrk-Mak/Ick axis engages in the turnaround process and subsequent retrograde IFT. These results suggest that ciliopathies defective in these IFT processes can be restored by the activation of the CCRK-MAK/ICK signaling pathway. Ick was proposed to induce disassembly of anterograde IFT trains at ciliary tips ([27](ref), [33](ref)), which is a critical step to promote the IFT turnaround process and subsequent retrograde transport. In this study, we showed that Mak, similar to Ick, regulates the IFT turnaround step, suggesting that Mak also induces the disassembly of anterograde IFT trains at ciliary tips. Because cytoplasmic dynein is probably incompletely inhibited by Ciliobrevin D treatment and *Dync2li1* knockdown, we hypothesized that IFT turnaround process promoted by Ick and Mak activation might be able to compensate for the defects in retrograde IFT caused by Ciliobrevin D treatment and *Dync2li1* knockdown. Although the precise functional relationship between the Ccrk-Mak/Ick pathway and IFT-A/cytoplasmic dynein-2 remains unclear, mutations in the genes encoding IFT-A and cytoplasmic dynein-2, and *ICK* are associated with a shared human ciliopathy, short rib-polydactyly syndrome ([93](ref)). Further mechanistic studies would reveal the extent to which our proposed therapeutic approach can be more generally applied to treat ciliopathies.

## Materials and Methods

### Animal care

All procedures conformed to the ARVO Statement for the Use of Animals in Ophthalmic and Vision Research, were approved by the Institutional Safety Committee on Recombinant DNA Experiments (approval ID 04913) and the Animal Experimental Committees of the Institute for Protein Research (approval ID R04-02-0), Osaka University, and were performed in compliance with the institutional guidelines. Mice were housed in a temperature-controlled room at 22°C with a 12-h light/dark cycle. Fresh water and rodent diet were available at all times. All animal experiments were performed with mice of either sex.

### Mouse lines

To generate the *Ccrk* flox mouse line, we subcloned an ~ 11.5-kb *Ccrk* genomic fragment using C57BL/6 genomic DNA by PCR, inserted one loxP site into intron 2 and another loxP site into intron 4, cloned it into a modified pBluescript II KS (+) vector (Agilent) to make a targeting construct, and transfected the linearized targeting construct into the JM8A3 embryonic stem cell line ([94](ref)). Culture, electroporation, and selection of JM8A3 cells were performed as described previously ([66](ref)). Embryonic stem cells heterozygous for targeted gene disruption were microinjected into C57BL/6 blastocysts to obtain chimeric mice. These chimeric mice were bred with C57BL/6 mice to obtain their progenies, which were subsequently crossed with B6-Tg(CAG-FLPe)37 mice (#RBRC01835; RIKEN BRC) to remove the flippase recognition target–flanked neo cassette using flippase (Flp) recombinase. We obtained $Ccrk^{+/-}$ mice by crossing the *Ccrk* flox mouse line with the female *BAC-Dkk3-Cre* transgenic mouse line, whose progenies exhibit complete recombination in all tissues, irrespective of the co-transmission of the *BAC-Dkk3-Cre* transgene ([50](ref)). Primer sequences used for genotyping are listed in Table S1. *Ick* flox mice ([33](ref)), $Ick^{-/-}$ mice ([33](ref)), $Cdkl5^{-/Y}$ mice ([95](ref)), $Mak^{-/-}$ mice ([96](ref)), and *Crx-CreERT2* transgenic mice ([66](ref)) have been described previously.

### Tamoxifen treatment

Tamoxifen (Toronto Research Chemicals) was dissolved in sunflower seed oil (Sigma-Aldrich) to a concentration of 20 mg/ml and injected 4 mg of it intraperitoneally into mice at 1 mo. The injected mice were analyzed at 2 mo.

### RT–PCR analysis

RT–PCR analysis was performed as described previously ([97](ref)). Total RNAs were extracted using TRIzol (Ambion) from tissues dissected from ICR mice at 4 wk, retinas from the control and *Ccrk* CKO mice at P14, and retinas from the control and *Ccrk* iCKO mice at 2 mo. Total RNA (2 μg) was reverse-transcribed into cDNA with random hexamers using the PrimeScript II reagent (TaKaRa). Human cDNA was purchased from Clontech. The cDNAs were used for PCR with rTaq polymerase (TaKaRa). The primer sequences used for amplification are listed in Table S1.

### In situ hybridization

In situ hybridization was performed as described previously ([98](ref)). Mouse eye cups were fixed using 4% PFA in PBS for 30 min at room temperature. The tissues were equilibrated in 30% sucrose in PBS overnight at 4°C, embedded in Tissue-Tek OCT compound 4583 (Sakura), and frozen. Digoxigenin-labeled antisense riboprobes for mouse *Ccrk* were synthesized by in vitro transcription using 11-digoxigenin UTPs (Roche). A *Ccrk* cDNA fragment for an in situ hybridization probe was generated by PCR, using mouse testicular

cDNA as a template. The primer sequences used for amplification are listed in Table S1.

## Plasmid constructs

Plasmids expressing EGFP, FLAG- or HA-tagged mouse Mak and Ick, FLAG-tagged EGFP, mouse IFT57, IFT88, IFT140, BBS8, and Ick-KD, and HA-tagged human ICK were previously constructed (33, 52, 99). Full-length cDNA fragments of mouse *Cdkl1*, *Cdkl2*, *Cdkl3*, *Cdkl4*, *Cdkl5*, *Gsk3b*, and *Dync2li1* were amplified by PCR using mouse retinal cDNA as a template and subcloned into the pCAGGSII-3xFLAG and/or pCAGGSII-2xHA (67, 100) vectors. Full-length cDNA fragments of mouse *Mok*, *Gsk3a*, and *Ccrk* were amplified by PCR using mouse testicular cDNA as a template and subcloned into the pCAGGSII-3xFLAG and/or pCAGGSII (52) vectors. A full-length cDNA fragment of human *MAK* was amplified by PCR using human retinal cDNA (Clontech) as a template and subcloned into the pCAGGSII-2xHA vector (67). The mouse Ick T157A mutation was introduced via site-directed mutagenesis using PCR. For shRNA-mediated *Dync2li1* knockdown, Dync2li1-shRNA and Control-shRNA cassettes were subcloned into the pBAsi-mU6 vector (Takara Bio). The target sequences were as follows: Dync2li1-shRNA2, 5′-GCTTTGTGGCA-CATTACTACG-3′; Dync2li1-shRNA3, 5′-GCAGGACTGGATTCTTTATGT-3′; Dync2li1-shRNA5, 5′-GGGAATTAATTGACCCATTTC-3′; Dync2li1-shRNA6, 5′-GCAAGTCAGAAGCTCTGTTAC-3′; and Control-shRNA, 5′-GACGTCTAACGGATTCGAGCT-3′ (101). To produce AAV–Rhodopsin kinase promoter–3xFLAG-Ick, a full-length cDNA fragment of mouse *Ick* was amplified by PCR using pCAGGSII-2xHA-mouse Ick as a template and subcloned into the pBluescript II KS (+)-3xFLAG vector, which was modified from the pBluescript II KS (+) vector. *3xFLAG-Ick* digested from pBluescript II KS (+)-3xFLAG-Ick was ligated into the pAAV-RK-IZsGreen vector (a kind gift from Dr. T Li) (58). The primer sequences used for the amplification are listed in Table S1.

## Chemicals

BGJ398 was purchased from ChemScene. A smoothened agonist (SAG) was obtained from Calbiochem. Ciliobrevin D was purchased from Millipore.

## Cell culture and transfection

HEK293T and NIH3T3 cells were cultured in DMEM (Sigma-Aldrich) containing 10% FBS and calf serum, respectively, supplemented with penicillin (100 $\mu$g/ml) and streptomycin (100 $\mu$g/ml) at 37°C with 5% $CO_2$. $Ick^{-/-}$ MEFs were derived from embryonic day 13.5 embryos and cultured in DMEM containing 10% FBS supplemented with penicillin (100 $\mu$g/ml) and streptomycin (100 $\mu$g/ml) at 37°C with 5% $CO_2$. Transfection was carried out using the calcium phosphate method for HEK293T cells and Lipofectamine LTX (Invitrogen) or Lipofectamine 3000 (Invitrogen) for NIH3T3 cells and $Ick^{-/-}$ MEFs. To induce ciliogenesis in transfected cells, the medium was replaced with serum-free medium 24 h after transfection, and the cells were cultured for 24 h in serum-free medium. NIH3T3 cells were incubated with 100 nM BGJ398, 100 nM SAG, or 10 $\mu$M Ciliobrevin D for 24 h in serum-free medium.

## Immunofluorescence analysis of cells and retinal sections

Immunofluorescence analysis of the cells and retinal sections was performed as described previously (102, 103). Cells were washed with PBS, fixed with 4% PFA in PBS for 5 min at room temperature, subsequently incubated with blocking buffer (5% normal donkey serum and 0.1% or 0.5% Triton X-100 in PBS) for 30 min at room temperature, and then immunostained with primary antibodies in blocking buffer overnight at 4°C. Cells were washed with PBS and incubated with secondary antibodies and DAPI (1:1,000; Nacalai Tesque) in blocking buffer for 2 h at room temperature. After washing three times with PBS, the samples were coverslipped with gelvatol. The mouse eyes or eye cups were fixed with 4% PFA in PBS for 15 s to 30 min at room temperature. Eye samples were rinsed in PBS, embedded in Tissue-Tek OCT compound 4583 (Sakura), frozen, and sectioned. The eye cup samples were rinsed in PBS, followed by cryoprotection using 30% sucrose in PBS overnight at 4°C, embedded in Tissue-Tek OCT compound 4583, frozen, and sectioned. Frozen 14- or 20-$\mu$m sections on slides were dried overnight at room temperature, rehydrated in PBS for 5 min, and incubated with blocking buffer for 30 min and then with primary antibodies overnight at 4°C. Slides were washed with PBS three times for 5 min each and incubated with fluorescent dye–conjugated secondary antibodies and DAPI for 2 h at room temperature while shielded from light. After washing three times with PBS, the sections were coverslipped with gelvatol. The specimens were observed under a laser confocal microscope (LSM700, LSM710, or LSM900; Carl Zeiss). Primary antibodies used in this study were as follows: mouse anti-Actub (1:1,000 or 1:2,000, 6-11B-1; Sigma-Aldrich), guinea pig anti-Ick (1:100) (33), rabbit anti-IFT88 (1:500, 13967-1-AP; Proteintech), rabbit anti-IFT140 (1:500, a kind gift from Dr. GJ Pazour) (104), rabbit anti-FLAG (1:1,000, F7425; Sigma-Aldrich), mouse anti-FLAG-M2 (F1804, 1:1,000; Sigma-Aldrich), mouse anti-$\gamma$-tubulin (1:250, T6557; Sigma-Aldrich), rabbit anti-pericentrin (1:1,000, ab4448; Abcam), guinea pig anti-Mak (1:1,000) (52), mouse anti-Cep164 (1:200, sc-515403; Santa Cruz), rabbit anti-Rhodopsin (1:2,500, LB-5597; LSL), goat anti-S-opsin (1:500, sc-14363; Santa Cruz or 1:100, 600-101-MP7S; ROCKLAND), rabbit anti-M-opsin (1:500, AB5405; Millipore), and rabbit anti-(pThr$^{674}$) Kif3a (1:250) (33). Cy3-conjugated (1:500; Jackson ImmunoResearch Laboratories) and Alexa Fluor 488–conjugated (1:500; Sigma-Aldrich) secondary antibodies were used. Ciliary length was measured using ZEN imaging software (ZEN [blue edition]; Carl Zeiss). The rod outer segment lengths and immunofluorescence signal intensities were measured and quantified using National Institutes of Health (NIH) ImageJ software. The Rhodopsin signals in the inner part of the photoreceptors were normalized to the total (OS + inner part) signals for Rhodopsin.

## Toluidine blue staining

Toluidine blue staining of the retinal sections was performed as described previously (105). The retinal sections were rinsed with PBS and stained with 0.1% toluidine blue in PBS for 1 min. After washing with PBS, the slides were covered with coverslips and immediately observed under a microscope. Retinal thickness was

measured and quantified using NIH ImageJ software as described previously ([99]).

## Transmission electron microscope analysis

Transmission electron microscopy was performed as previously described ([106]). Mouse eye cups were fixed with 2% glutaraldehyde and 2% PFA in PBS for 30 min. Retinas were fixed with 1% osmium tetroxide for 90 min, dehydrated through a graded series of ethanol (50%–100%), cleared with propylene oxide, and embedded in epoxy resin. Sections were cut on an ultramicrotome (Ultracut E; Reichert-Jung), stained with 2% uranyl acetate and Sato's lead staining solution ([107]), and observed under a transmission electron microscope (H-7500; Hitachi Co).

## ERG recording

ERGs were recorded as described previously ([108]). Briefly, mice were dark-adapted overnight and then anesthetized with an intraperitoneal injection of 100 mg/kg ketamine and 10 mg/kg xylazine diluted in saline (Otsuka). The pupils were dilated using topical 0.5% tropicamide and 0.5% phenylephrine HCl. The ERG responses were measured using a PuREC system with LED electrodes (Mayo Corporation). Mice were placed on a heating pad and stimulated with an LED flash. Four levels of stimulus intensities ranging from −4.0 to 1.0 log cd s m$^{-2}$ were used for the scotopic ERGs. After the mice were light-adapted for 10 min, the photopic ERGs were recorded on a rod-suppressing white background of 1.5 log cd m$^{-2}$. Four levels of stimulus intensities ranging from −0.5 to 1.0 log cd s m$^{-2}$ were used for the photopic ERGs. Eight responses at −4.0 log cd s m$^{-2}$ and four responses at −3.0 log cd s m$^{-2}$ were averaged for the scotopic recordings. Sixteen responses were averaged for the photopic recordings.

## Western blot analysis

Western blot analysis was performed as described previously ([109]). HEK293T cells were washed twice with TBS and lysed in SDS sample buffer or a lysis buffer supplemented with protease inhibitors (Buffer A: 20 mM Tris–HCl, pH 7.4, 150 mM NaCl, 1% NP-40, 1 mM EDTA, 1 mM PMSF, 2 μg/ml leupeptin, 5 μg/ml aprotinin, and 3 μg/ml pepstatin A). Mouse retinas were lysed in buffer A or B (20 mM Tris–HCl, pH 7.4, 150 mM NaCl, and 1% NP-40 supplemented with phosphatase inhibitor cocktail [Roche]). Samples were resolved by SDS–PAGE and transferred to PVDF membranes (Millipore) using a semidry transfer cell (Bio-Rad) or iBlot system (Invitrogen). The membranes were blocked with blocking buffer (3% skim milk or 1% BSA, and 0.05% Tween-20 in TBS) and incubated with primary antibodies overnight at 4°C. The membranes were washed with 0.05% Tween-20 in TBS three times for 10 min each and then incubated with secondary antibodies for 2 h at room temperature. The signals were detected using Chemi-Lumi One L (Nacalai) or Pierce Western Blotting Substrate Plus (Thermo Fisher Scientific). The following primary antibodies were used: rabbit anti-Kif3a (1: 1,500, ab11259; Abcam), mouse anti-FLAG M2 (1:5,000 or 1:10,000, F1804; Sigma-Aldrich), rabbit anti-GFP (1:2,500, 598; MBL), rabbit anti-pMak (1:1,000, PA5-105526; Invitrogen), guinea pig anti-Mak (1:150) ([52]), guinea pig anti-Ccrk (1:500, generated in this study), and mouse anti-α-tubulin (DM1A, 1:5,000, T9026; Cell Signaling). The following secondary antibodies were used: horseradish peroxidase–conjugated anti-mouse IgG (1:10,000; Zymed), anti-guinea pig IgG (1:10,000; Jackson Laboratory), and anti-rabbit IgG (1:10,000; Jackson Laboratory). Phosphorylated Kif3a was detected by 6% SDS–PAGE with 20 μM Phos-tag acrylamide (FUJIFILM Wako), according to the manufacturer's instruction ([110], [111]). The band intensity was quantified using NIH ImageJ software.

## Antibody production

Antibody production was performed as described previously ([112]). Briefly, a cDNA fragment encoding the C-terminal portion of mouse Ccrk (residues 289–346) was amplified by PCR and subcloned into the pGEX-4T-3 vector (GE Healthcare). The GST-fused Ccrk protein was expressed in *Escherichia coli* strain BL21 (DE3) and purified using glutathione Sepharose 4B (GE Healthcare). An antibody against Ccrk was obtained by immunizing guinea pigs with purified GST-fused Ccrk.

## Analysis of scRNA-seq data

To analyze the publicly available scRNA-seq data of the adult human retina ([57]), the processed data (ae_exp_proc_all.tsv) obtained from http://www.ebi.ac.uk/arrayexpress/experiments/E-MTAB-7316 were imported into the R package Seurat (version 4.3.0) ([113]). The data were normalized using the LogNormalize method with a scale factor of 10,000, and the variable features were identified using FindVariableFeatures with 2,000 genes. Dimensionality reduction was then performed using PCA. Cell clustering was performed using FindNeighbors and FindClusters (dim = 20, resolution = 0.6). UMAP was performed to visualize cell clusters using the RunUMAP function. Cell identities were assigned to each cluster as described previously ([57]).

## AAV production and subretinal injection

AAV production and subretinal injection of AAVs were performed as previously described ([114]). AAVs were produced by triple transfection of an AAV vector plasmid, adenovirus helper plasmid, and AAV helper plasmid (pAAV2-8 SEED) into AAV-293 cells using the calcium phosphate method. The cells were harvested 72 h after transfection and lysed using four freeze–thaw cycles. The supernatant was collected by centrifugation and treated with Benzonase nuclease (Novagen). The viruses were purified using an iodixanol gradient. The gradient was formed in ultra-clear centrifuge tubes (14 × 95 mm, Beckman) by first adding 54% iodixanol (Axis-Shield) in PBS-MK buffer (1×PBS, 1 mM MgCl$_2$, and 25 mM KCl), followed by overlaying 40% iodixanol in PBS-MK buffer, 25% iodixanol in PBS-MK buffer containing phenol red, and 15% iodixanol in PBS-MK buffer containing 1 M NaCl. The tubes were centrifuged at 284,100$g$ using an SW40Ti rotor. The 54% to 40% fraction containing the virus was collected using an 18-gauge needle and concentrated using an Amicon Ultra Centrifugal Filter Ultracel-100K (Millipore). The titer of each AAV (in vector genomes [VG]/ml) was determined by qPCR using SYBR GreenER qPCR Super Mix (Invitrogen) and Thermal

Cycler Dice Real Time System Single MRQ TP700 (Takara) according to the manufacturer's protocols. Quantification was performed using Thermal Cycler Dice Real Time System software (Takara). The primers used for AAV titrations are listed in Table S1. The titer of AAV-RK-N-FLAGx3-Ick was adjusted to ~1.8 × 10$^{12}$ VG/ml. 0.3 $\mu$l of the AAV preparation was injected into the subretinal spaces of P1 $Mak^{-/-}$ mice. The injected retinas were harvested at P14.

### Drug administration

BGJ398 dissolved in DMSO (2 or 4 mg/ml) was diluted in 3.5 mM HCl and 5% DMSO. Two mg/kg of BGJ398 was administered to mice every day by subcutaneous injection.

### Luciferase reporter assay

Reporter gene assays were performed using Nano-Glo Dual-Luciferase Reporter Assay System (Promega) according to the manufacturer's protocol. To generate the reporter construct, a minimal promoter (5'-AGACACTAGAGGGTATATAATGGAAGCTCGACTTCCAG-3') and 8x Gli1-binding sites (115) were cloned into the pGL3-Basic vector (Promega), in which *Firefly luciferase* was replaced with *NanoLuc luciferase* amplified from the pNLF-N [CMV/Hygro] vector (Promega) by PCR. The pGL3-Control vector (Promega) was co-transfected to normalize the transfection efficiency. Plasmids expressing Dync2li1-shRNA2, Dync2li1-shRNA3, or Dync2li1-shRNA6, and Ick, Mak iso1, Mak iso2, or Ccrk were transfected with the reporter constructs into NIH3T3 cells. After 24 h of transfection, cells were incubated in serum-free medium containing 100 nM SAG for 24 h and washed with PBS. Luminescence signals were detected using GloMax Multi + Detection System (Promega).

### Sequence alignment and structure modeling

Structure-based sequence alignment of mouse Mak and Ick was performed using Clustal Omega (116) and ESPript (117). Mouse Mak and Ick 3D structures were obtained from the AlphaFold Protein Structure Database developed by DeepMind, in which the protein structure was predicted based on the amino acid sequence (https://alphafold.ebi.ac.uk/) (118, 119). The predicted 3D structures of the kinase domains (residues 4–284) in Mak and Ick proteins were visualized and aligned using PyMOL.

### Statistical analysis

Data are represented as the mean ± SD. Statistical analysis was performed using an unpaired *t* test, one-way ANOVA, or two-way ANOVA, as indicated in the figure legends. A value of *P* < 0.05 was taken to be statistically significant.

## Supplementary Information

## Acknowledgements

We thank Y Shinkai for the $Mak^{-/-}$ mouse, GJ Pazour for the anti-IFT140 antibody, T Li for the pAAV-RK-IZsGreen vector, and M Kadowaki, A Tani, A Ishimaru, T Tsujii, M Wakabayashi, Y Nakamura, Y Kinooka, K Yoshida, S Okochi, K Fukunaga, S Kubo, H Yamamoto, Y Sugita, LR Varner, R Saito, K Kobayashi, T Yasui, M Nishi, K Wada, T Takeda, R Shimizu, S Zhou, and K Nakamura for technical assistance. This work was supported by Grant-in-Aid for Scientific Research (21H02657, 24K09996) and Grant-in-Aid for Challenging Research (Exploratory) (23K18199) from the Japan Society for the Promotion of Science, AMED-CREST (21gm1510006) from the Japan Agency for Medical Research and Development, Japan Science and Technology Agency (JST) Moonshot R&D (JPMJMS2024), JST COI-NEXT (JPMJPF2018), the Takeda Science Foundation, and the Uehara Memorial Foundation.

### Author Contributions

T Chaya: conceptualization, data curation, formal analysis, funding acquisition, validation, investigation, visualization, methodology, project administration, and writing—original draft, review, and editing.
Y Maeda: formal analysis, validation, investigation, visualization, methodology, and writing—original draft, review, and editing.
R Tsutsumi: formal analysis, validation, investigation, and visualization.
M Ando: validation, investigation, and visualization.
Y Ma: validation, investigation, and visualization.
N Kajimura: investigation.
T Tanaka: resources.
T Furukawa: conceptualization, supervision, funding acquisition, methodology, project administration, and writing—review and editing.

### Conflict of Interest Statement

T Chaya and T Furukawa are inventors on a patent application related to this work filed by the Osaka University.

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
