## [Reviewer comments · Life Science Alliance]

Life Science Alliance

Ccrk-Mak/Ick signaling is a ciliary transport regulator essential for retinal photoreceptor survival

Taro Chaya, Yamato Maeda, Ryotaro Tsutsumi, Makoto Ando, Yujie Ma, Naoko Kajimura, Teruyuki Tanaka, and Takahisa Furukawa

DOI: <https://doi.org/10.26508/lsa.202402880>

Corresponding author(s): Takahisa Furukawa, Osaka University

Review Timeline:

Submission Date:	2024-06-08
Editorial Decision:	2024-07-25
Revision Received:	2024-08-09
Editorial Decision:	2024-08-26
Revision Received:	2024-08-27
Accepted:	2024-08-27

Transaction Report:

July 25, 2024

Re: Life Science Alliance manuscript #LSA-2024-02880-T

Prof. Takahisa Furukawa
Institute for Protein Research, Osaka University
Laboratory for Molecular and Developmental Biology
3-2 Yamadaoka
Suita 565-0871
Japan

Dear Dr. Furukawa,

Thank you for submitting your manuscript entitled "Ccrk-Mak/Ick kinase signaling axis is a ciliary transport regulator essential for retinal photoreceptor maintenance" to Life Science Alliance. The manuscript was assessed by expert reviewers, whose comments are appended to this letter. We invite you to submit a revised manuscript addressing the Reviewer comments.

Thank you for this interesting contribution to Life Science Alliance. We are looking forward to receiving your revised manuscript.

Sincerely,

B. MANUSCRIPT ORGANIZATION AND FORMATTING:

Reviewer #1 (Comments to the Authors (Required)):

This paper reports studies of two ciliary tip localized kinases, Mak and Ick, and an associated upstream activator, Ccrk kinase. The authors propose that these kinases act cooperatively to form a network that contributes to build and maintain mammalian photoreceptors via their effect on intraflagellar transport. This conclusion is based on a large amount of work using perturbation and rescue of kinase function in various combinations to examine the effects on ciliary morphology. The authors further propose that targeting this kinase network may be important for understanding and treating retinal diseases such as retinitis pigmentosa.

While I do not question the scientific rigor of the work presented in this paper, after reading it I felt ambivalent about recommending its publication in this journal. Some of my concerns are listed below:

1. One reason is that I think that the paper represents only an incremental advance in our understanding of the fundamental mechanism and regulation of intraflagellar transport and ciliogenesis - there are many studies that have previously illuminated roles of ciliary kinases in IFT (some of which are listed in the bibliography). However, it is quite possible that the paper would be better suited to a more medically oriented journal, given the authors' proposal about its relevance to retinal pathology, this is something I am not qualified to judge.

2. In relation to the above I strongly disagree with the sentiment that "the regulatory mechanisms of IFT and their physiological roles are poorly understood". IFT is quite a mature field by now and understanding of its mechanism and physiological roles are quite extensive (see cited reviews) and the regulation of the process has been covered in an excellent review by Mul W, Mitra A, Peterman EJG. Mechanisms of Regulation in Intraflagellar Transport. *Cells*. 2022 Sep 2;11(17):2737. Unfortunately, I don't see the current study as fundamentally advancing what we already know, although this is only one opinion.

3. Perhaps part of the problem is that the paper is long and difficult to read. If the authors were able to cut the paper into a more succinct version which focuses on the major points and explains clearly what the major advances are, it would probably be received more favorably, at least by this reviewer?

4. Although the presentation, including the title, focus on photoreceptor cilia, many of the experiments are done on cultured cells, whose primary cilia appear very different from the convoluted membranous-associated axonemes of photoreceptor cells. This is not well-justified in the presentation of the paper.

5. Actually, in relation to the above point, sensory cilia that also have similarly complex membrane-bounded axonemes to vertebrate photoreceptors, as well as the presence of extensive distal singlet MT domains associated with ciliary-signaling, are found in the *C. elegans* nervous system. It might be useful to address this point and the general nature of system-specific variability in IFT and its control at the end of the second paragraph of the introduction by saying something like; "There is significant system-specific variability in the details of IFT, for example, in contrast to *Chlamydomonas* and some mammalian cilia, two distinct anterograde motors operate in *C. elegans* cilia (Prevo et al, 2015, *Nat Cell Biol*) and the return of kinesin-2 motors from the ciliary tip to the base occurs through diffusion in *Chlamydomonas* (Chien et al., 2017; Pedersen et al., 2006), by retrograde transport in *C. elegans*, and by both processes in mammalian cells (Luo et al., 2017; Prevo et al., 2015; Williams et al., 2014)." Similarly the turnaround mechanism also appears different in different systems depending upon whether only one or two distinct anterograde motors are involved (see Prevo et al, 2015 and Zhang Z, et al. Direct imaging of intraflagellar-transport turnarounds reveals that motors detach, diffuse, and reattach to opposite-direction trains. *Proc Natl Acad Sci U S A*. 2021;118(45). These variations appear very relevant to the current study.

6. The sections on ciliary defects caused by cytoplasmic dynein inhibition are not convincing to me - is this due to inhibition of cytoplasmic dynein-1 which may transport essential cargo from the cytoplasm to the base of the cilium/transition zone or is it due to the inhibition of the retrograde IFT motor, cytoplasmic dynein-2?

Reviewer #2 (Comments to the Authors (Required)):

Taro Chaya et al.

This is a systematic analysis of the function of MAK (male germ cell-associated kinase regulating ciliary length), ICK (official symbol Cilk1, ciliogenesis associated kinase 1, enabling phosphorylation at Ser and Thr), CDKL5 (cyclin-dependent kinase-like 5) and CCRK (Ccrk kinase, an upstream activator of MAK) using cell lines and conditional knockouts in mouse. Authors show that MAK is a ciliary tip-localized IFT regulator that cooperatively acts with the ciliopathy kinase ICK. Mutations in MAK are associated with RP62 and mutations in ICK with epilepsy.

- ICK plays a minor role in the regulation of IFT in retinal photoreceptor cells. ICK DKO shows reduced scotopic and photopic ERG responses at 6M of age.

- Mak/Ick dKO in mouse resulted in loss of photoreceptor axonemes and retina degeneration.

- MAK plays a major role in photoreceptor IFT

- Mak overexpression in cultured cells restored ciliary defects caused by Ick deficiency.

- CDKL5 is dispensable for retinal photoreceptor function and maintenance

- Gene delivery of Ick and pharmacological inhibition of FGF receptors ameliorated retinal degeneration in Mak^{-/-} mice.

- Ccrk kinase is an upstream activator of Mak and Ick in retinal photoreceptor cells.

- Authors propose that the CCRK-MAK/ICK axis functions as an IFT regulator crucial for retinal photoreceptor maintenance.

The paper is well written and molecular genetics of cell lines and mouse knockouts are state of the art. Comparison of results in primary cilia and mouse photoreceptors are interesting and useful, particularly differences between invertebrates and photoreceptors.

Specific points

Results, parag. 1. "We found Ick signals in the distal regions of the photoreceptor ciliary axonemes in the control retina but not in the Ick CKO retina (Fig. S1A). However, Fig. S1A still shows traces on ICK in the inner segment and ONL. .

Page 9, center: "...photoreceptor ciliary axonemes were elongated in the Mak^{-/-} retina compared to those in the Mak^{+/+} retina (Fig. 1G)." 1G shows IFT88 and IFT140 responses in Mak^{+/+} and Mak^{-/-} rods. In Mak^{+/+} IFT88 and IFT140 locate to both ends of the CC. AcTub is at CC. in Mak^{-/-}, CC seems extended into the proximal axoneme but IFT88 is NOT at the tip of the axoneme. The length of the axoneme is unclear.

Page 15, bottom: "These results demonstrate that Ick and Mak activation can rescue ciliary defects caused by cytoplasmic dynein inhibition." But how? Removing or dissociating the inhibitor from dynein? Phosphorylation of dynein light chains? Do the authors have an idea how this may happen?

Fig. 2C DAPI control P14 should be replaced. It shows nuclei in the OS, this makes no sense. Similar effects are shown in 6A.

Response to reviewers:**Reviewer #1:**

This paper reports studies of two ciliary tip localized kinases, Mak and Ick, and an associated upstream activator, Ccrk kinase. The authors propose that these kinases act cooperatively to form a network that contributes to build and maintain mammalian photoreceptors via their effect on intraflagellar transport. This conclusion is based on a large amount of work using perturbation and rescue of kinase function in various combinations to examine the effects on ciliary morphology. The authors further propose that targeting this kinase network may be important for understanding and treating retinal diseases such as retinitis pigmentosa.

While I do not question the scientific rigor of the work presented in this paper, after reading it I felt ambivalent about recommending its publication in this journal. Some of my concerns are listed below:

Thank you for reviewing our manuscript. We appreciate your understanding of our paper and sincerely considered your comments. Accordingly, we have prepared a new version of the manuscript.

1. One reason is that I think that the paper represents only an incremental advance in our understanding of the fundamental mechanism and regulation of intraflagellar transport and ciliogenesis - there are many studies that have previously illuminated roles of ciliary kinases in IFT (some of which are listed in the bibliography). However, it is quite possible that the paper would be better suited to a more medically oriented journal, given the authors' proposal about its relevance to retinal pathology, this is something I am not qualified to judge.

Thank you for your comments. "AIMS & SCOPE" of *Life Science Alliance* (<https://www.life-science-alliance.org/about-journal>) mentions that "*Life Science*

Alliance welcomes submission in the following biological processes” with a list of multiple biological processes. We believe that the current manuscript fits for “Cell biology”, “Development”, “Genetics, Gene Therapy, & Genetic Disease”, “Medical Research” and “Neuroscience” among the listed biological processes, and is suitable for publication in *Life Science Alliance*.

2. In relation to the above I strongly disagree with the sentiment that "the regulatory mechanisms of IFT and their physiological roles are poorly understood". IFT is quite a mature field by now and understanding of its mechanism and physiological roles are quite extensive (see cited reviews) and the regulation of the process has been covered in an excellent review by Mul W, Mitra A, Peterman EJG. Mechanisms of Regulation in Intraflagellar Transport. Cells. 2022 Sep 2;11(17):2737. Unfortunately, I don't see the current study as fundamentally advancing what we already know, although this is only one opinion.

According to the reviewer’s comment, we have modified the text in the Abstract as follows.

(Page 3, line 3 - page 3, line 5)

The formation, function, and maintenance of primary cilia depend crucially on intraflagellar transport (IFT); however, the regulatory mechanisms of IFT at ciliary tips are poorly understood.

This modified text reflects the description in the Introduction “At the tip of the cilia, IFT trains disassemble and reassemble for turnaround and retrograde transport (26); however, the underlying regulatory mechanisms remain poorly understood (27)” (Page 5, line 7 - page 5, line 9).

3. Perhaps part of the problem is that the paper is long and difficult to read. If the authors were able to cut the paper into a more succinct version which focuses on the

major points and explains clearly what the major advances are, it would probably be received more favorably, at least by this reviewer?

Thank you for your comment. If possible, we also would like to make our manuscript more concise. We believe that our conclusion is supported by the current results as mentioned “This conclusion is based on a large amount of work using perturbation and rescue of kinase function in various combinations to examine the effects on ciliary morphology.” in the first paragraph by the reviewer. We believe that all current results data are not omittable to support the conclusion. If more specifically designated, we may shorten a possibly non-essential part of the Introduction and/or Discussion.

4. Although the presentation, including the title, focus on photoreceptor cilia, many of the experiments are done on cultured cells, whose primary cilia appear very different from the convoluted membranous-associated axonemes of photoreceptor cells. This is not well-justified in the presentation of the paper.

According to the reviewer’s comment, we have modified the text as follows.

(Page 4, line 6 - page 4, line 10)

For example, retinal photoreceptor cells possess a light-sensory structure containing components of the phototransduction cascade, including Rhodopsin and cone opsins, the outer segments, which are the specialized primary cilia that have the convoluted membranous-associated axonemes but share common structural features with those of primary cilia in other cell types (4).

5. Actually, in relation to the above point, sensory cilia that also have similarly complex membrane-bounded axonemes to vertebrate photoreceptors, as well as the presence of extensive distal singlet MT domains associated with ciliary-signaling, are found in the C. elegans nervous system. It might be useful to address this point and the general nature of system-specific variability in IFT and its control at the end of the second paragraph of the introduction by saying something like; " There is significant

system-specific variability in the details of IFT, for example, in contrast to Chlamydomonas and some mammalian cilia, two distinct anterograde motors operate in C. elegans cilia (Prevo et al, 2015, Nat Cell Biol) and the return of kinesin-2 motors from the ciliary tip to the base occurs through diffusion in Chlamydomonas (Chien et al., 2017; Pedersen et al., 2006), by retrograde transport in C. elegans, and by both processes in mammalian cells (Luo et al., 2017; Prevo et al., 2015; Williams et al., 2014)." Similarly the turnaround mechanism also appears different in different systems depending upon whether only one or two distinct anterograde motors are involved (see Prevo et al, 2015 and Zhang Z, et al. Direct imaging of intraflagellar-transport turnarounds reveals that motors detach, diffuse, and reattach to opposite-direction trains. Proc Natl Acad Sci U S A. 2021;118(45). These variations appear very relevant to the current study.

According to the reviewer's comment, we have added the following text at the end of the second paragraph of the Introduction:

(Page 5, line 10 - page 5, line 5 from the bottom)

Sensory cilia that have complex membrane-bounded axonemes similar to vertebrate photoreceptors, as well as the presence of extensive distal singlet microtubule domains associated with ciliary signaling, are found in the *Caenorhabditis elegans* (*C. elegans*) nervous system. There is significant system-specific variability in the details of IFT, for example, in contrast to *Chlamydomonas reinhardtii* (*Chlamydomonas*) and some mammalian cilia, two distinct anterograde motors operate in *C. elegans* cilia (28) and the return of kinesin-2 motors from the ciliary tip to the base occurs through diffusion in *Chlamydomonas* (26,29), by retrograde transport in *C. elegans*, and by both processes in mammalian cells (28,30,31). Similarly, the turnaround mechanism also appears to be different in different systems depending on whether only one or two distinct anterograde motors are involved (28,32).

6. The sections on ciliary defects caused by cytoplasmic dynein inhibition are not convincing to me - is this due to inhibition of cytoplasmic dynein-1 which may transport

essential cargo from the cytoplasm to the base of the cilium/transition zone or is it due to the inhibition of the retrograde IFT motor, cytoplasmic dynein-2?

We knocked down *Dync2li1*, a gene encoding cytoplasmic dynein-2 light intermediate chain 1, in NIH3T3 cells and observed ciliary defects (Fig. 4M-S, 6D-F, and S6K-M), indicating that the observed ciliary defects were due to the inhibition of the retrograde IFT motor cytoplasmic dynein-2.

Reviewer #2:

Taro Chaya et al.

This a systematic analysis of the function of MAK (male germ cell-associated kinase regulating ciliary length), ICK (official symbol Cilk1, ciliogenesis associated kinase 1, enabling phosphorylation at Ser and Thr), CDKL5 (cyclin-dependent kinase-like 5) and CCRK (Ccrk kinase, an upstream activator of MAK) using cell lines and conditional knockouts in mouse. Authors show that MAK is a ciliary tip-localized IFT regulator that cooperatively acts with the ciliopathy kinase ICK. Mutations in MAK are associated with RP62 and mutations in ICK with epilepsy.

- ICK plays a minor role in the regulation of IFT in retinal photoreceptor cells. ICK DKO shows reduced scotopic and photopic ERG responses at 6M of age.*
- Mak/Ick dKO in mouse resulted in loss of photoreceptor axonemes and retina degeneration.*
- MAK plays a major role in photoreceptor IFT*
- Mak overexpression in cultured cells restored ciliary defects caused by Ick deficiency.*
- CDKL5 is dispensable for retinal photoreceptor function and maintenance*
- Gene delivery of Ick and pharmacological inhibition of FGF receptors ameliorated retinal degeneration in Mak^{-/-} mice.*
- Ccrk kinase is an upstream activator of Mak and Ick in retinal photoreceptor cells.*
- Authors propose that the CCRK-MAK/ICK axis functions as an IFT regulator crucial for retinal photoreceptor maintenance.*

The paper is well written and molecular genetics of cell lines and mouse knockouts are state of the art. Comparison of results in primary cilia and mouse photoreceptors are interesting and useful, particularly differences between invertebrates and photoreceptors.

Thank you for reviewing our manuscript. We appreciate your understanding of our paper and sincerely considered your comments. We prepared a new version of the manuscript accordingly.

Specific points

Results, parag. 1. "We found Ick signals in the distal regions of the photoreceptor ciliary axonemes in the control retina but not in the Ick CKO retina a (Fig. S1A). However, Fig. S1A still shows traces on ICK in the inner segment and ONL.

According to the reviewer's comment, we added the text as follows:

(Page 8, line 10 - page 8, line 12)

We also found signals stained with the anti-Ick antibody in the inner segments and outer nuclear layer (ONL) in both control and *Ick* CKO retinas (Fig. S1A), suggesting that these signals were non-specific.

Page 9, center: "...photoreceptor ciliary axonemes were elongated in the Mak^{-/-} retina compared to those in the Mak^{+/+} retina (Fig. 1G)." 1G shows IFT88 and IFT140 responses in Mak^{+/+} and Mak^{-/-} rods. In Mak^{+/+} IFT88 and IFT140 locate to both ends of the CC. AcTub is at CC. in Mak^{-/-}, CC seems extended into the proximal axoneme but IFT88 is NOT at the tip of the axoneme. The length of the axoneme is unclear.

According to the reviewer's comment, we modified the text as follows:

(Page 11, line 11 - page 11, line 8 from the bottom)

As observed in our previous study (52), photoreceptor connecting cilia were elongated in the *Mak^{-/-}* retina compared to those in the *Mak^{+/+}* retina (Fig. 1G). In contrast to the *Ick* CKO retinas (Fig. S1B), we observed that both IFT88 and IFT140 were concentrated at the tips of photoreceptor connecting cilia in the *Mak^{-/-}* retina compared with those in the *Mak^{+/+}* retina (Fig. 1G), which is similar to our previous observation that IFT components are concentrated at the ciliary tips in *Ick^{-/-}* mouse embryonic fibroblasts (MEFs) (33).

Page 15, bottom: "These results demonstrate that Ick and Mak activation can rescue ciliary defects caused by cytoplasmic dynein inhibition." But how? Removing or dissociating the inhibitor from dynein? Phosphorylation of dynein light chains? Do the authors have an idea how this may happen?

Thank you for your comment. Ick was proposed to induce disassembly of anterograde IFT trains at ciliary tips (27,33), which is a critical step to promote IFT turnaround process and subsequent retrograde transport. In the current study, we showed that Mak, similar to Ick, regulates the IFT turnaround step, suggesting that Mak also induces the disassembly of anterograde IFT trains at ciliary tips. Since cytoplasmic dynein is probably incompletely inhibited by Ciliobrevin D treatment and *Dync2li1* knockdown, we hypothesized that IFT turnaround process promoted by Ick and Mak activation might be able to compensate for the defects in retrograde IFT caused by Ciliobrevin D treatment and *Dync2li1* knockdown. We added descriptions to the Discussion as follows:

(Page 28, line 10 - page 28, line 6 from the bottom)

Ick was proposed to induce disassembly of anterograde IFT trains at ciliary tips (27,33), which is a critical step to promote IFT turnaround process and subsequent retrograde transport. In this study, we showed that Mak, similar to Ick, regulates the IFT turnaround step, suggesting that Mak also induces the disassembly of anterograde IFT trains at ciliary tips. Since cytoplasmic dynein is probably incompletely inhibited by Ciliobrevin D treatment and *Dync2li1* knockdown, we hypothesized that IFT turnaround process promoted by Ick and Mak activation might be able to compensate for the defects in retrograde IFT caused by Ciliobrevin D treatment and *Dync2li1* knockdown.

27. Nachury MV (2022) The gymnastics of intraflagellar transport complexes keeps trains running inside cilia. *Cell* 185: 4863-4865. doi:10.1016/j.cell.2022.12.005
33. Chaya T, Omori Y, Kuwahara R, Furukawa T (2014) Ick is essential for cell type-specific ciliogenesis and the regulation of ciliary transport. *EMBO J* 33: 1227-1242. doi:10.1002/embj.201488175

Fig. 2C DAPI control P14 should be replaced. It shows nuclei in the OS, this makes no sense. Similar effects are shown in 6A.

In accordance with the reviewer's comment, we have replaced the old images with new images in Fig. 2C and 6A, as shown below. The new images are indicated by red rectangles.

Figure 2

Figure 6

August 26, 2024

RE: Life Science Alliance Manuscript #LSA-2024-02880-TR

Prof. Takahisa Furukawa
Osaka University
Laboratory for Molecular and Developmental Biology
3-2 Yamadaoka
Suita 565-0871
Japan

Dear Dr. Furukawa,

Thank you for submitting your revised manuscript entitled "Ccrk-Mak/Ick signaling is a ciliary transport regulator essential for retinal photoreceptor survival". We would be happy to publish your paper in Life Science Alliance pending final revisions necessary to meet our formatting guidelines.

- please be sure that the authorship listing and order is correct
- please add the Twitter handle of your host institute/organization as well as your own or/and one of the authors in our system

A. FINAL FILES:

B. MANUSCRIPT ORGANIZATION AND FORMATTING:

Thank you for your attention to these final processing requirements. Please revise and format the manuscript and upload materials within 5 days.

Sincerely,

Reviewer #2 (Comments to the Authors (Required)):

This a systematic analysis of the function of MAK (male germ cell-associated kinase regulating ciliary length), ICK (official symbol CilK1, ciliogenesis associated kinase 1, enabling phosphorylation at Ser and Thr), CDKL5 (cyclin-dependent kinase-like 5) and CCRK (Ccrk kinase, an upstream activator of MAK) using cell lines and conditional knockouts in mouse. Authors show that MAK is a ciliary tip-localized IFT regulator that cooperatively acts with the ciliopathy kinase ICK. Mutations in MAK are associated with RP62 and mutations in ICK with epilepsy.

The paper is well written and molecular genetics of cell lines and mouse knockouts are state of the art. Comparison of results in primary cilia and mouse photoreceptors are interesting and useful, particularly differences between invertebrates and photoreceptors.

Authors have responded well to comments from reviewers. This reviewer has no further comments.

August 27, 2024

RE: Life Science Alliance Manuscript #LSA-2024-02880-TRR

Prof. Takahisa Furukawa
Osaka University
Laboratory for Molecular and Developmental Biology
3-2 Yamadaoka
Suita 565-0871
Japan

Dear Dr. Furukawa,

Thank you for submitting your Research Article entitled "Ccrk-Mak/Ick signaling is a ciliary transport regulator essential for retinal photoreceptor survival". It is a pleasure to let you know that your manuscript is now accepted for publication in Life Science Alliance. Congratulations on this interesting work.

DISTRIBUTION OF MATERIALS:

Again, congratulations on a very nice paper. I hope you found the review process to be constructive and are pleased with how the manuscript was handled editorially. We look forward to future exciting submissions from your lab.

Sincerely,
